# A robust multivariate structure of interindividual covariation between psychosocial characteristics and arousal responses to visual narratives

Jinyoung Kim[1], Eunseong Bae[2], Yeonhwa Kim[1], Chae Young Lim[3], Ji-Won Hur[4], Jun Soo Kwon[1,5], Sang-Hun Lee[1]*

1 Department of Brain and Cognitive Sciences, Seoul National University, Seoul, Republic of Korea, 2 Department of Statistics, University of California, Davis, California, United States of America, 3 Department of Statistics, Seoul National University, Seoul, Republic of Korea, 4 Department of Psychology, Korea University, Seoul, Republic of Korea, 5 Department of Psychiatry, Seoul National University College of Medicine, Seoul, Republic of Korea

* visionsl@snu.ac.kr

**Data Availability Statement:** Our data is now publicly available at the following URL: https://osf.io/a97hp.

## Abstract

People experience the same event but do not feel the same way. Such individual differences in emotion response are believed to be far greater than those in any other mental functions. Thus, to understand what makes people individuals, it is important to identify the systematic structures of individual differences in emotion response and elucidate how such structures relate to what aspects of psychological characteristics. Reflecting this importance, many studies have attempted to relate emotions to psychological characteristics such as personality traits, psychosocial states, and pathological symptoms across individuals. However, systematic and global structures that govern the across-individual covariation between the domain of emotion responses and that of psychological characteristics have been rarely explored previously, which limits our understanding of the relationship between individual differences in emotion response and psychological characteristics. To overcome this limitation, we acquired high-dimensional data sets in both emotion-response (8 measures) and psychological-characteristic (68 measures) domains from the same pool of individuals (86 undergraduate or graduate students) and carried out the canonical correlation analysis in conjunction with the principal component analysis on those data sets. For each participant, the emotion-response measures were quantified by regressing affective-rating responses to visual narrative stimuli onto the across-participant average responses to those stimuli, while the psychological-characteristic measures were acquired from 19 different psychometric questionnaires grounded in personality, psychosocial-factor, and clinical-problem taxonomies. We found a single robust mode of population covariation, particularly between the 'accuracy' and 'sensitivity' measures of arousal responses in the emotion domain and many 'psychosocial' measures in the psychological-characteristics domain. This mode of covariation suggests that individuals characterized with positive social assets tend to show polarized arousal responses to life events.

**Funding:** JK and SL were supported by the Brain Research Programs and Basic Research Laboratory Program through the National Research Foundation of Korea (NRF) funded by the Ministry of Science and ICT of South Korea (https://english.msit.go.kr). Grant numbers are NRF-2017M3C7A1047860 for the Brain Research Programs and NRF-2018R1A4A1025891 for the Basic Research Laboratory Program. The funders had no role in study design, data collection and analysis, decision to publish, or preparation of the manuscript.

**Competing interests:** The authors have declared that no competing interests exist.

## Introduction

Some emotions, especially six [1–3] or up to nine [4–6] categorical emotions which are tightly associated with distinct nonverbal expressions, appear to be universal at high degrees. On the other hand, there has been evidence also showing that perception of emotion varies across cultures at significant degrees [7, 8]. Furthermore, even within the same culture, individuals substantively differ in their emotion perception or reactivity, exhibiting different responses (e.g., from being mildly surprised, to shivering, and even to crying) to the same object or event (e.g., a scene in a horror movie).

The appraisal processes [9, 10], multi-component compositions [11, 12], or unfolding dynamics [10, 13] of emotion have been suggested to be among many factors contributing to such individual differences [14]. Specifically, the appraisal process is based on an individual's idiosyncratic life-long history of experiences, making emotional experience highly subjective. The multi-component nature of emotion implies that the same category of emotion (e.g., anger) may differ in actual composition across individuals, which leads to finely nuanced variations in emotional experience [15]. The literature on emotion dynamics indicates that individuals substantively differ in emotion duration [16] and in the variability of intensity over time [17, 18].

Not surprisingly, considering these factors contributing to individual differences in emotion, individual differences in emotion are far greater than those in any other mental functions [19, 20], on which the field of personality research is grounded: "emotions make people individuals", as often phrased [14]. In this sense, i.e., to understand what makes people individuals, it is crucial to identify the systematic structures of individual differences in emotion and elucidate how such structures relate to what aspects of psychological characteristics including long-lasting personality traits, psychosocial states, and psychiatric symptoms. Advancing such identification and elucidation would not just advance our understanding of the idiosyncratic nature of human emotion but also provide practical solutions to the problems in diverse social or clinical situations. For instance, effective pedagogical strategies can be tailored to individuals based on individual differences in emotion to enhance the communication between tutors and students [21], or identification of emotion perception styles that are associated with certain mental disorders on social networks can be used as an indirect yet highly natural probe for detecting high-susceptible individuals from a normal population [22].

Reflecting this importance, a large volume of work has been carried out to relate emotions to psychological characteristics across individuals. These studies can be sorted in terms of what aspects of emotion were probed, namely 'accuracy', 'bias', 'variability', and 'differentiability'. Previous studies that probed the accuracy of emotion [23–29], which refers to how less deviated an individual's emotion responses are from the population norm, reported its association with personality traits or mental problems. For instance, individuals with extraversion and neuroticism traits tended to be high and low, respectively, in emotion accuracy [28]. Clinical problems, such as depression, anxiety, and schizophrenia, typically showed negative associations with emotion accuracy [23–26]. Previous studies on the bias of emotion [30–32], which refers to how biased an individual's emotion responses are toward a certain category (e.g., 'happy') or affective state (e.g., 'positive side on the valence axis'), reported its associations with psychosocial factors or personality traits. For instance, individuals with good coping skills tended to be positively biased in valence [32], or those with openness traits were positively biased both in valence and arousal [31]. The variability of emotion refers to how inconsistent an individual's emotion responses to the same event or highly similar events are [11, 33] and has been reported to show a positive relationship with neuroticism and a negative relationship with agreeableness [11]. Lastly, the differentiability of emotion (also dubbed "emotion

granularity") refers to how finely an individual can discriminate emotional situations—how sensitive an individual is in detecting subtle nuances or differences between emotional situations [34–36]. Individuals with high differentiability of emotion are reported to exhibit high scores of self-esteem [37].

We note that the previous studies mentioned above were mostly designed to test theories or hypotheses about the relationship between certain aspects of emotion and psychological characterics relying on pairwise comparison analysis methods. Even when multiple emotion measures and different psychological characteristics were collected within single studies, only bivariate relationships were mostly inspected [38, 39]. This 'hypothesis-driven and regional' approach taken by these previous studies may efficiently address their respective 'regional' questions by testing the predictions of interest. Despite this merit, the 'hypothesis-driven and regional' approach may not be ideal for revealing the systematic and global structure that governs the across-participant covariation between the domain of emotion responses and that of psychological characteristics, especially when considering the aforementioned factors contributing to large individual differences in emotion. Specifically, the hypothesis-driven and regional approach may be insensitive to the presence or absence of a potential structure that can be defined only in a multidimensional space of emotion or psychological characteristics. In other words, a significant pairwise correlation does not warrant its participation in the true global structure that governs the relationship between the two domains [40, 41]. Likewise, an insignificant pairwise correlation does not necessarily mean that it does not contribute to the true global structure [42].

The goal of the current work is to identify the systematic and global structure that governs the across-individual covariation between the domain of emotion and that of psychological characteristics. To achieve this goal effectively, we took the 'data-driven and global' approach—as an alternative to the hypothesis-driven and regional approach—and considered several other important aspects, as follows. First, we used visual narrative stimuli, 15-second long film excerpts of various genres, to probe emotion responses. Visual narratives can be considered ideal for promoting the across-individual variability because they contain the aforementioned ingredients contributing to individual differences of emotion: various affective states are expected to be unfolded [13, 43] as people undergo the appraisal process [9, 10] by integrating multiple cues under complex and natural contexts [11, 12]. Second, we collected as many and diverse measures as possible in both domains. As for the domain of psychological characteristics, we collected a total of 68 measures using 19 different batteries of psychometric questionnaires, which cover the subdomains including 'personality', 'psychosocial factors', and 'clinical problems'. As for the domain of emotion, we acquired two-dimensional ('arousal' and 'valence') affective-state responses to visual narratives and derived the four measures that have been reported by the previous work to be associated with certain psychological characteristics (we named the measures as 'accuracy', 'bias', 'consistency', and 'sensitivity'). Lastly, we carried out multivariate analyses to discover a global structure of the across-individual covariation between the emotion-response and psychological-characteristics domains. In doing so, to address the known limitations of multivariate analysis methods associated with the dimensionality and interpretability issues, we conducted the canonical correlation analysis (CCA) [44] in conjunction with the principal component analysis (PCA) [45], which allowed us to effectively search a compact and interpretable feature space for significant across-individual covariations between specific styles of emotion responses and particular profiles of psychological characteristics.

To anticipate results, the multivariate analyses on the data collected from 86 individuals revealed a single robust mode of covariation that links the domains of emotion responses and psychological characteristics. Specifically, the 'accuracy' and 'sensitivity' measures of arousal responses in the emotion domain and many 'psychosocial-factor' measures in the psychological-

characteristics domain contributed to the mode of population covariation. Based on further analyses on those measures with significant contributions, we reached an interpretation that the mode reflects the tendency of individuals characterized with positive social perspectives to show polarized arousal responses to life events.

## Methods

### Participants

We recruited 86 Korean undergraduate students of similar ages (41 females, $M_{age}$ = 21.4, age range: 18–24 years; Table 1). We justified the sample size by proceeding with simulations for power and specificity (see S1 Appendix for detail). All participants were interviewed by trained clinicians to be prescreened for neurological and/or psychiatric disorders. Six participants who had high (moderate to severe) Beck Depression Inventory (BDI) or Beck Anxiety Inventory (BAI) scores were excluded from further analysis. Participants all had a normal or corrected-to-normal vision. In addition, participants were also cataloged for their sex, age, IQ, and family income to statistically de-confound the individual differences that might potentially confound the relationship between psychological characteristics and emotion responses. This study was approved by the Seoul National University Research Ethics Committee, and informed written consent was obtained from all participants prior to actual participation.

### Psychological-characteristic measures

To acquire a comprehensive and unbiased set of psychological characteristics, we used a total of 19 psychometric questionnaires that were associated with diverse taxonomies that capture

**Table 1. Demographic summary of participants.**

| Demographic variables | | Frequency (n) | Percentage (%) |
|---|---|---|---|
| **Sex** | Male | 45 | 52.3 |
| | female | 41 | 47.7 |
| **Age** | 18–20 | 28 | 32.6 |
| | 20–24 | 58 | 67.4 |
| **Monthly household income** | less than $ 1,000 | 2 | 2.3 |
| | $ 1,000-$ 1,999 | 2 | 2.3 |
| | $ 2,000-$ 2,999 | 10 | 11.6 |
| | $ 3,000-$ 3,999 | 11 | 12.8 |
| | $ 4,000-$ 4,999 | 13 | 15.1 |
| | $ 5,000-$ 5,999 | 12 | 14.0 |
| | $ 6,000-$ 6,999 | 9 | 10.5 |
| | $ 7,000-$ 7,999 | 3 | 3.5 |
| | $ 8,000-$ 8,999 | 8 | 9.3 |
| | $ 9,000-$ 9,999 | 3 | 3.5 |
| | $ 10,000 or more | 10 | 11.6 |
| | Not stated | 3 | 3.5 |
| **IQ** | 91–100 | 2 | 2.3 |
| | 101–110 | 13 | 15.1 |
| | 111–120 | 36 | 41.9 |
| | 121–130 | 30 | 34.9 |
| | 131–140 | 5 | 5.8 |
| **Total** | | 86 | 100.0 |

individual differences. These taxonomies included: (i) the personality taxonomies that capture individual differences in relatively enduring behavioral tendencies based on the 'Big Five' model [46, 47], the 'Reinforcement Sensitivity' theory [48, 49], and Cloninger's 'Psychobiological model of temperament and character' [50]; (ii) the psychosocial-factor taxonomies that capture individual differences in capacity for recovery after significant adversity [51], in-person social support [52, 53], perceived social rank [54], capacity for empathy [55], perceived quality of life [56], recent life experience [57], and attitude towards themselves [58]; (iii) the clinical-problem taxonomies that capture individual differences in propensity for major mental problems such as psychological personality disorders [59], anxiety disorders [60, 61], substance abuse [62, 63], suicidal thinking [64], depression [65], and affective disorders [66]. A full list of the psychological-characteristic measures and psychometric questionnaires is available in the supporting information (S1–S3 Tables). All the questionnaires were provided in Korean, being translated if needed.

Since it took roughly 5 hours to complete the entire questionnaires, participants brought the questionnaires their home and turned in their answers a week later. Considering the cognitive burden on participants [67], we instructed them to fill in the questionnaires over multiple days by taking breaks of sufficient length. To prevent the incompletion of the questionnaires [68], we checked whether there exist any missed or inappropriate responses to items upon reception of the questionnaires and, if so, asked participants to respond to such items on site.

## Visual narrative stimuli

One of the authors (Y.K.), a film expert who majored in film art and worked in film-editing companies, built the library of visual narratives (VNs) by referring to the Internet Movie Database (IMDb) and Schaefer and his colleagues' work [69] relying on the following guidelines. First, the referred video sources were diverse and balanced in the genre, including action adventure, biography, comedy, documentary, drama, family, fantasy, mystery, horror, romance, sci-fi, sport, thriller, and western genres. Second, the scenes were selected such that they collectively covered a wide range of affective states, both in the valence and arousal dimensions. Third, every excerpt consisted of events that constituted a coherent piece of storytelling, such that it *could be* readily described with a few sentences. The last guideline was considered to ensure that an affective state was induced as a 'visual story' unfolds for each clip.

Most of the VN stimuli were excerpted from motion pictures (130 clips from 124 different motion pictures). Some affective states (e.g., states of low arousal and neutral valence) rarely occur in the motion-picture database that we referred to. To cover such affective states, music videos (7 clips from 4 music videos) or TV commercials (7 clips from 4 TV commercials) were also referred to. There is no specific reason to believe that these non-motion-picture clips differ from the motion-picture clips in the effectiveness of inducing affective states because both types of clips induce affective states with visual stories in the same manner. Notwithstanding, we confirmed that the emotion measures acquired using only the 130 motion-picture clips were highly correlated with those using the entire clips (S1 Fig, panel B). More details of the VN stimuli are provided in the supporting information (S1 File).

All VN stimuli were edited to be 15-second long, which is close to the length of commercial ads on TV or the Internet. They were made soundless to focus on nonverbal emotion perception and standardized in size (1400-pixel width, 744-pixel height), temporal frequency (24 frames per second), and color format (8-bit RGB). Stimulus presentation and collection of participants' responses were controlled using Psychophysics Toolbox extensions [70–72] in conjunction with MATLAB 2014b on an iMac computer with OS X (Apple Inc.).

## Emotion rating task

On each trial, participants were asked to indicate their emotional states after viewing freely (and without fixation) a 15-second VN stimulus displayed on the computer screen. Emotion ratings were collected for the dimensions of valence and arousal using the 9-point self-assessment manikin scale (SAM) [73]. Participants were given as much time as they required to rate stimuli before submitting their final ratings to the computer system. The rating session consisted of 6 blocks of 24 trials, and the order of 144 VN stimuli was randomized across participants. Four practice trials were completed prior to the main task to ensure that participants understood the instructions. The data from these practice trials were excluded from the analysis.

The across-participant averages of the affective states assigned to the VN stimuli (Fig 1A) were widely distributed over both dimensions of the affective space, exhibiting a typical 'V-shape' pattern—valence scores tend to bifurcate toward the negative and positive poles as arousal scores increase–, which has been repeatedly reported in previous work [74, 75]. We also expected that our VN stimuli would induce substantial individual variability in emotion responses. Indeed, the rating scores for the same VN stimulus varied considerably between individuals (Fig 1B), which resulted in standard deviations ranging from 0.63 to 1.88 in the valence dimension and from 1.10 to 2.02 in the arousal dimension.

## Emotion-response measures

Having confirmed that our VN stimuli covered the affective space in a representative manner while inducing sufficient individual differences (Fig 1A and 1B), we quantified those individual differences with a set of 'emotion-response measures', which measures how much the emotion rating patterns of individuals deviate from the 'normative' pattern in several aspects. Here, the 'normative' pattern refers to the distribution of emotion ratings averaged across all participants for the entire library of VN stimuli (Fig 1A). This population average can be considered as the 'typical' emotion responses that are shared across participants and thus represent empirical approximations of the 'normative' affective states induced by the VN stimuli.

Specifically, for a given individual $i$, the normative response was calculated as the average across the entire population except for the individual $i$. We then linearly regressed the participant $i$'s response to a visual narrative $l$, $r^{i,l}$, onto the normative ratings, $r^l_{norm}$, over the 144 VN stimuli, using the following regression model:

$$r^{i,l} = \alpha^i + \beta^i r^l_{norm} + \varepsilon^{i,l},$$

where $\varepsilon^{i,l}$ is the error, which was minimized to estimate the intercept, $\alpha^i$, and slope, $\beta^i$. To get the best-unbiased estimators of regression coefficients, the regression model was fit using the method of weighted least squares rather than ordinary least squares, because VN stimuli differed substantially in across-participant variability (standard deviations ranged from 1.10 to 2.02 for arousal ratings and from 0.63 to 1.88 for valence ratings). As a result, squared residual errors were weighted by the reciprocals of variances [76]. After fitting the regression model, we computed the proportion of the variance of $r^{i,l}$ explained by the linear regression onto $r^l_{norm}$, $\delta^i$. This triplet of regression parameters, $\{\alpha^i, \beta^i, \delta^i\}$, provides a complementary set of distinct aspects that reflect how an individual's emotion responses deviate from the normative responses, as follows: $\alpha^i$ reflects the extent to which a given individual $i$'s emotion responses are biased, providing a 'bias' measure; $\beta^i$ reflects how sensitively a given individual $i$'s emotion responses change as the normative responses change, providing a 'sensitivity' measure; $\delta^i$ reflects how noisy or unpredictable a given individual $i$'s emotion responses are, providing a

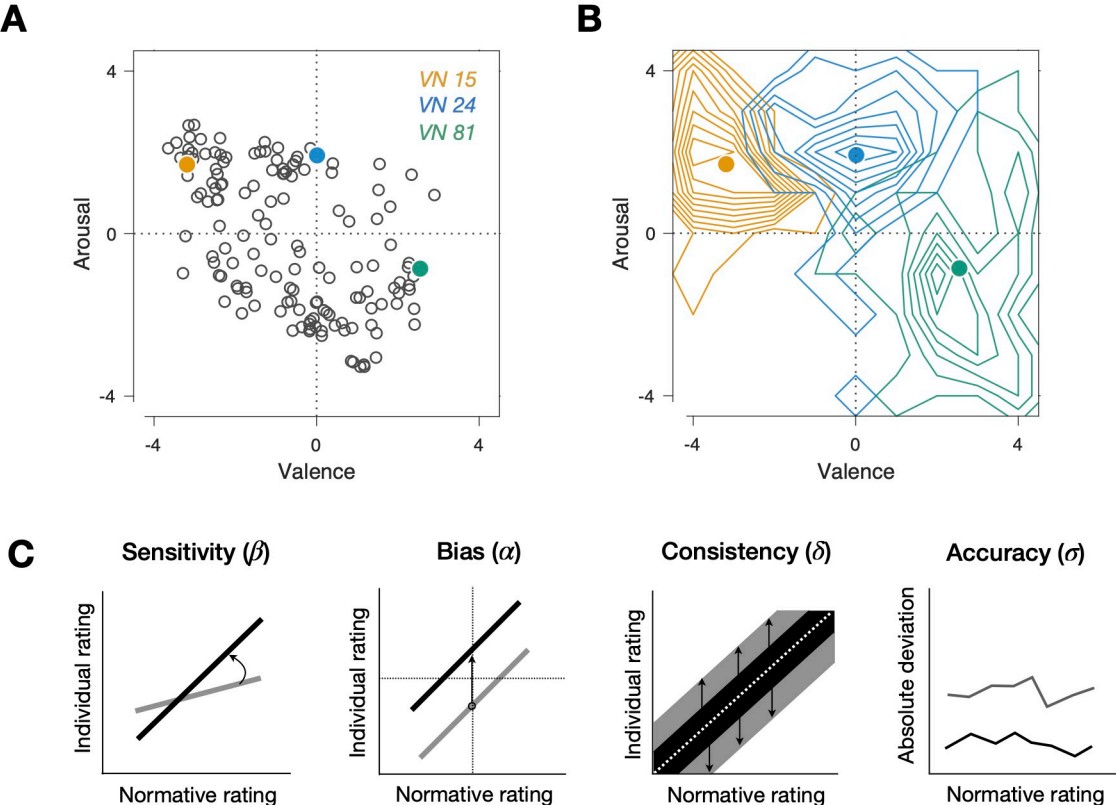

**Fig 1. The procedure of acquiring emotion responses and defining emotion measures.** (A-B) Emotion-rating responses to visual narratives (VNs). (A) Normative (i.e., averaged across participants) rating responses to the 144 VN stimuli plotted in the affective-state space of valence and arousal. The colored dots are the normative responses to the three example VN stimuli (VN 15, VN 24, and VN 81). (B) Contour histograms of individual emotion ratings of the three example VN stimuli. The colored dots are the same as the corresponding ones in A. (C) Definition of the emotion measures. Left three panels: the 'bias', 'sensitivity', and 'consistency' measures were defined by regressing individual participants' emotion rating responses onto the normative responses. Schematic examples of regression lines or confidence intervals are shown for individuals with high (black) and low (gray) scores of the 'bias' (left), 'sensitivity' (middle), and 'consistency' (right) measures. Right panel: the 'accuracy' measures were defined as the mean of absolute deviations from the normative responses. Example vectors of the absolute deviations are shown for individuals with high (black) and low (gray) scores of the 'accuracy' measure.

'consistency' measure (Fig 1C). Besides these measures based on regression analysis, we calculated the accuracy measures that have widely been used by the previous studies on individual differences in emotion responses [27, 28, 77]. The 'accuracy' measure, $\sigma^i$, was defined as the mean of absolute deviations from the normative responses across all 144 VN stimuli. The sign was reversed (-1×mean of absolute deviation) so that higher values mean higher degrees of accuracy. Considering the possibility that individual differences may exist independently between the two dimensions, the regressions and the accuracy calculation were performed separately for the arousal and valence dimensions. In sum, the way a given individual assigns emotional states to the VN stimuli was described by a vector of eight measures, $\theta^i = \{\alpha_a^i, \beta_a^i, \delta_a^i, \sigma_a^i, \alpha_v^i, \beta_v^i, \delta_v^i, \sigma_v^i\}$, where the subscripts, $a$ and $v$ denote the two subdomains of emotion, 'arousal' and 'valence', respectively.

## Canonical Correlation Analysis (CCA)

To meet the prerequisites of CCA and to avoid redundancy, we preprocessed the raw psychological-characteristics data, an 80×68 (subject × individual-characteristics measures) matrix

$C_{R1}$, and the raw emotion data, an 80×8 (subject × emotion measures) matrix $E_{R1}$, before carrying out the CCA in the following procedure. First, the raw measures were screened for extreme distributions of values by applying two criteria, 'extreme homogeneity' and 'extreme outliers'. A distribution with 'extreme homogeneity' was defined as one in which more than 90% of participants had an identical single value, whereas a distribution with 'extreme outliers' was defined as one in which the average squared deviation of values from their median was smaller by 100-fold than the maximum squared deviation, as follows:

$$\max_{i} \; d^i > 100 \times mean(d);$$

$$d^i = \left[ \theta^i - median(\theta) \right]^2,$$

where $\theta^i$ refers to a participant $i$'s psychological-characteristics or emotion measure. Only one psychological-characteristics measure was excluded in this step, which resulted in $C_{R2}$(80×67 matrix) for the psychological-characteristics data and $E_{R2}$ (= $E_{R1}$) for the emotion-response data. Next, to avoid the unwanted effects of hidden outlier values and to satisfy the assumption of normal distribution, which is required of the CCA [78, 79], we rescaled the values of $C_{R2}$ and $E_{R2}$ into rank values and then 'Gaussianized' those rank-scaled values by mapping them onto the normalized value space [80], producing $C_R$ and $E_R$ (Fig 2A and 2B).

Next, the PCA was conducted on $C_R$ and $E_R$, which resulted in $C_P$ and $E_P$ (Fig 2A and 2B), to avoid the overfitting due to the high dimensionality of the psychological-characteristics measures and to orthogonalize the individual measures in $C_R$ and $E_R$. In building $C_P$, the number of PCs was varied from 8 to 30 because it may affect the results of CCA. This particular range of PC numbers was determined by applying two criteria: (i) more than 60% of the total variance of $C_R$ should be explained; (ii) the canonical correlation should be significant. As for $E_P$, the number of PCs was fixed to 8. The subjects-to-subjects covariance matrix was fed into eigenvalue decomposition to determine subject-wise eigenvectors with the largest eigenvalues for each measure type [78]. As a result, 100% of the total variance of $E_R$ was explained by $E_P$ while 61.35% (for 8 eigenvectors) to 92.18% (for 30 eigenvectors) of the total variance of $C_R$ was explained by $C_P$. Although the dimension of $E_R$ was low, we applied PCA to $E_R$ because we wanted to use the procedure identical to that used for $C_R$ (but the results remain almost unchanged whether PCA was applied to $E_R$ or not). To prevent potential confounds with socio-demographic factors, $C_P$ and $E_P$ were de-confounded for age, sex, IQ, and income scores prior to CCA. Specifically, those socio-demographic variables underwent a rank-based inverse normal transformation and were regressed out from both $C_P$ and $E_P$.

We conducted the CCA on $C_P$ and $E_P$ using the '*canoncorr*' function in the Statistics and Machine Learning Toolbox of MATLAB. The CCA initially identified an orthogonal set of 'pairs of canonical variates' ($\{(C_{M1}, E_{M1}), (C_{M2}, E_{M2}),\ldots,(C_{Mk}, E_{Ck}),\ldots,(C_{Mn}, E_{Mn})\}$) that maximizes the pairwise correlations between linear combinations of $C_P$ and $E_P$. The first CCA mode, ($C_{M1}$, $E_{M1}$), was defined as follows:

$$(C_{M1}, E_{M1}) = \left( C_P \cdot W_{M1}^C, \; E_P \cdot W_{M1}^E \right),$$

where

$$\left( W_{M1}^C, W_{M1}^E \right) = \max_{W_{M1}^C, W_{M1}^E} corr\left( C_P \cdot W_{M1}^C, \; E_P \cdot W_{M1}^E \right)$$

Similarly, the remaining CCA modes, $\{(C_{M2}, E_{M2}),\ldots,(C_{Mk}, E_{Mk}),\ldots,(C_{Mn}, E_{Mn})\}$ were sequentially defined by finding a pair of vectors, $W_{Mk}^C$ and $W_{Mk}^E$, which maximizes the correlation between paired variates, $C_{Mk}$ and $E_{Mk}$, with a constraint that these newly added variates,

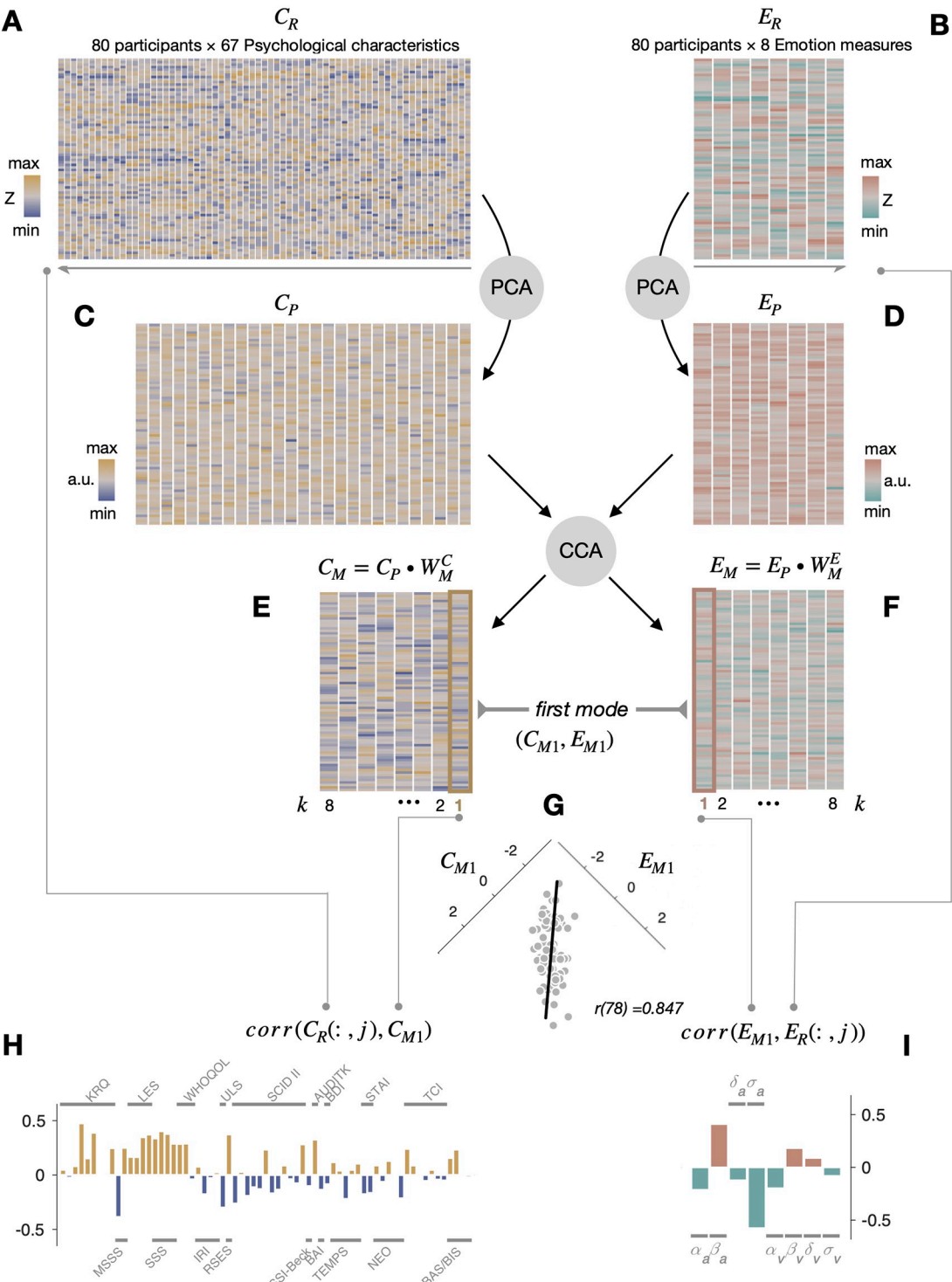

**Fig 2. The procedure of the multivariate analyses.** (A) A matrix of psychological-characteristics measures ($C_R$). Columns, 67 screened measures. Rows, 80 screened participants. The rows are identical among the matrices shown in A-F. The color hues and saturations of pixels correspond to the signs and strengths, respectively, of the normalized psychological-characteristics scores. (B) A matrix of emotion-response measures ($E_R$). Columns, 8 measures. The color hues and saturations correspond to the signs and strengths, respectively, of the normalized emotion-response measures. (C) A PCA-score matrix for the psychological-characteristics measures ($C_P$). Columns, 27 subject-wise eigenvectors with largest eigenvalues. In the actual analysis, the number of eigenvectors was varied from 8 to 30. (D) A PCA-score matrix for the emotion-response measures ($E_P$). Columns, eight subject-wise eigenvectors. (C, D) The color hues and saturations of pixels correspond to the signs and strengths, respectively, of the normalized PCA scores. (E) A CCA-score matrix of the psychological-characteristics measures ($C_{Mk}$). Columns, eight

canonical variates. (F) A CCA-score matrix of the emotion-response measures ($E_{Mk}$). Columns, eight canonical variates. (E, F) The color hues and saturations of pixels correspond to the signs and degrees, respectively, of the normalized CCA scores. (G) There was a strong and significant correlation between the canonical variates paired in the first (k = 1) CCA mode. The CCA scores of the first canonical psychological-characteristics variate ($C_{M1}$) are plotted against those of the first canonical emotion variate ($E_{M1}$) over individual participants (gray dots). A line is the linear regression of $E_{M1}$ onto $C_{M1}$. (H) Correlations of $C_{M1}$ with the individual columns of $C_R$ in A. The order of bars is identical to that of the columns in A. (I) Correlations of $E_{M1}$ with the individual columns of $E_R$ in B. The order of bars is identical to that of the columns in B.

$C_{Mk}$ and $E_{Mk}$, must be orthogonal (uncorrelated) to all the preceding modes (Fig 2E and 2F). To assess the statistical significances of the CCA modes, we permuted $C_P$ over participants 10,000 times and computed the correlation for each pair of the corresponding columns of $C_M$ and $E_M$.

As the final step, we computed the correlations of the paired canonical variates that constitute the significant first CCA mode, $C_{M1}$ and $E_{M1}$, with their raw measures, $C_R$ and $E_R$, respectively, to identify the specific psychological characteristics and the emotion-response measures that covary across participants via the first CCA mode.

## Results

### Distribution and reliability of the emotion-response measures

The across-participant distributions of the emotion-response measures are summarized in Table 2. For the measures of consistency ($\delta$) and accuracy ($\sigma$), the distribution means were greater in valence than in arousal ($\delta$: $t(158) = 3.62$, $p < .001$; $\sigma$: $t(158) = 4.75$, $p < 0.001$). By contrast, for all the measures except for sensitivity ($\beta$), the standard deviations of the distributions were greater in arousal than in valence ($\alpha$: $F(79,79) = 6.54$, $p < 0.001$; $\beta$: $F(79,79) = 1.41$, $p = 0.127$; $\delta$: $F(79,79) = 2.26$, $p < 0.001$; $\sigma$: $F(79,79) = 3.10$, $p < 0.001$).

We evaluated the reliability of the emotion-response measures in two aspects. First, when the trials were split into two subsets, such that two different sets of VN stimuli were used in those two subsets, the measures were highly consistent between those subsets (see S1 Fig for detailed procedures and results). Second, the emotion-response measures remain consistent even when the normative emotion responses (i.e., across-participant average responses) were defined from much a smaller number of subjects (see S2 Fig for detailed procedures and results).

### Distribution of the psychological-characteristics measures

We classified the psychological-characteristics measures into three groups, namely the 'psychosocial factors', 'clinical problems', and 'personality' measures, depending on the original purposes of the questionnaires (Fig 3A). For the purpose of screening out the measures with unhealthy across-participant distributions, we inspected whether a given distribution contains a few individuals with extremely outlying scores (Fig 3B) and whether it is too narrow for

**Table 2. Summary statistics of emotion measures.**

| Measures | Arousal | | Valence | |
|:---:|:---:|:---:|:---:|:---:|
| | $\mu$ | SD | $\mu$ | SD |
| Bias ($\alpha$) | 0.00 | 0.70 | 0.00 | 0.27 |
| Sensitivity ($\beta$) | 0.99 | 0.30 | 1.00 | 0.25 |
| Consistency ($\delta$) | 0.68 | 0.14 | 0.75 | 0.09 |
| Accuracy ($\sigma$) | -1.18 | 0.39 | -0.94 | 0.22 |

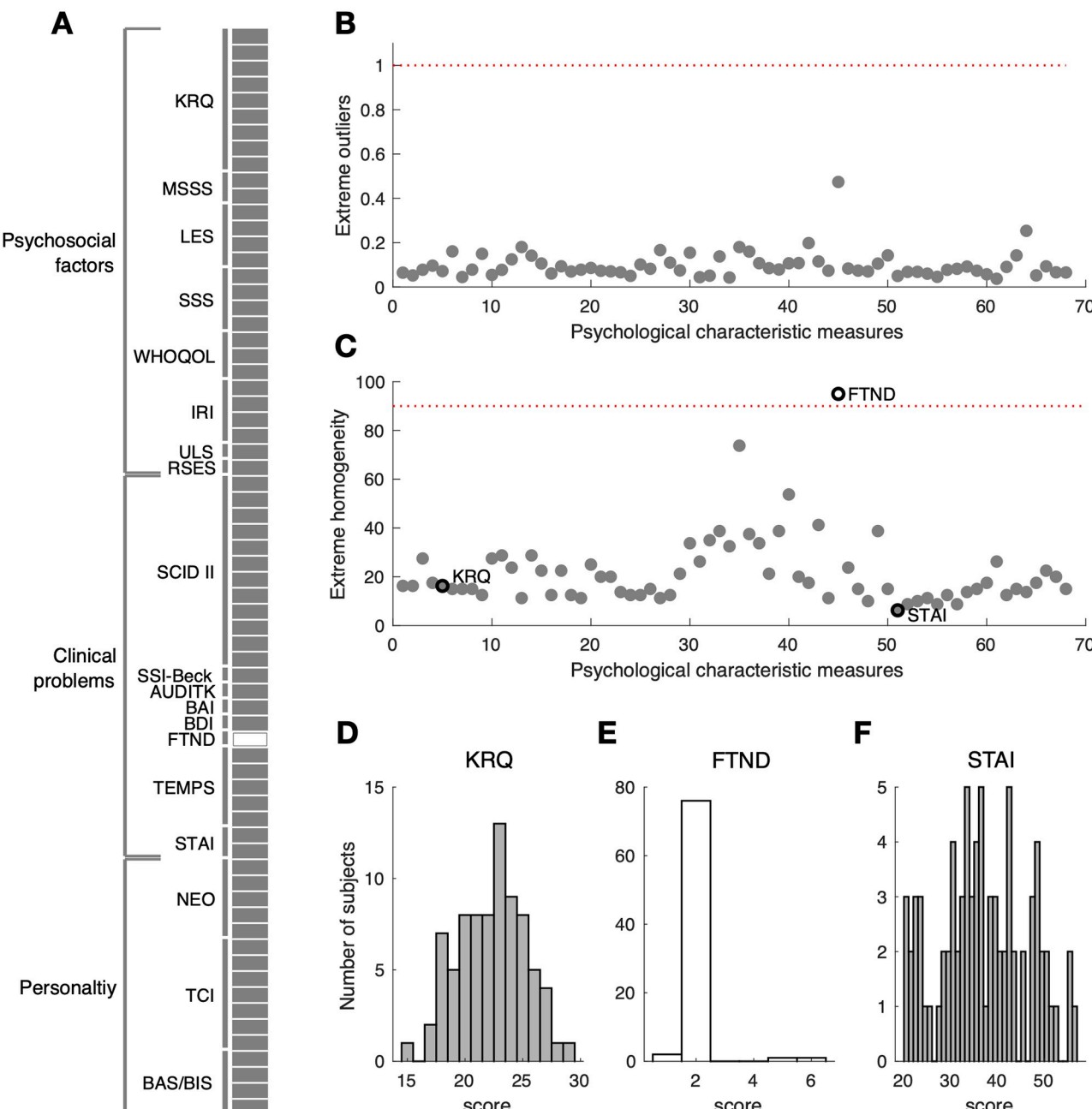

**Fig 3. Distribution analysis of the psychological-characteristics measures.** (A) The group identities and measure numbers of the 19 psychological-characteristics questionnaires. The questionnaires from which individual measures (horizontal bars) and the groups to which the questionnaires belong are indicated by the word labels with vertical bars and brackets, respectively. The measure that was not used for further analysis is indicated by the horizontal empty bar. (B,C) Results of the distribution analysis. (B) None of the measurements had extremely outlying scores, as indicated by the dots that all fell below the criterion (the red dotted line). (C) A single measure (FTND) had an extremely homogenous distribution, in which the majority (95%) of participants had the same score, as indicated by the dot located above the criterion (a red dotted line). (D-F) The distributions of three example measures. By comparing the distribution shapes and their corresponding scores in C, the exceptionally strong homogeneity of the FTND distribution can readily be appreciated. KRQ, Korean resilience quotient; MSSS, MacArthur scale of subjective social status; LES, life experiences survey; SSS, Social Support Scale; WHOQOL, world health organization quality of Life; IRI, interpersonal reactivity index; ULS, UCLA Loneliness Scale; RSES, Rosenberg self-esteem scale; SCID-II, structured clinical interview schedule for DSM-IV Axis-II disorder; SSI-Beck, Beck scale for suicidal ideation; AUDIT-K, Alcohol Use disorder identification test; BAI, Beck anxiety inventory; BDI, Beck depression inventory; FTND, Fagerström Test for Nicotine Dependence; TEMPS, temperament evaluation of Memphis, Pisa, Paris and San Diego; STAI, state-trait Anxiety Inventory; NEO, revised NEO personality inventory; TCI, temperament and character inventory; BAS/BAS, behavioral approach/inhibition system.

individuals to be distinguished from one another (Fig 3C). As a result, the FTND (Fagerström Test for Nicotine Dependence) measure was screened out because its distribution was extremely narrow (i.e., too homogenous because 95% of the participants were non-smokers; Fig 3E) compared to the remaining variables (the distributions of two example measures are shown in Fig 3D and 3F).

## The robustness of the first CCA mode

Regardless of the varying number of PC components that used to define the psychological-characteristics input to CCA ($C_P$; see Methods for the rationale for choosing the 23 different PC numbers), only the first CCA mode remains significant (permutation-test results are shown in Table 3; see Methods for the detailed procedure of permutation tests). In what follows, given this robustness of the first mode, we assessed the contributions of the raw measures (i.e., the individual columns of $C_M$ and $E_M$) to the population covariation between the psychological-characteristics and emotion-response domains based on the correlations between the canonical variables of the first CCA mode ($C_{M1}$ and $E_{M1}$) and the raw measures ($C_R$ and $E_R$), as graphically illustrated in Fig 2H and 2I.

## The psychological-characteristics measures contributing to the first CCA mode

To identify the psychological-characteristics measures contributing significantly to the CCA mode, we tested the significance of the correlation of the first-mode psychological-

**Table 3. Correlation coefficient and permutation test result of the first CCA mode.**

| Number of principal components | | $corr(C_{M1}, E_{M1})$ | $p$ |
|---|---|---|---|
| Emotion measures | Psychological characteristics | | |
| 8 | 8 | 0.69 | 0.002 |
| | 9 | 0.70 | 0.002 |
| | 10 | 0.70 | 0.004 |
| | 11 | 0.72 | 0.003 |
| | 12 | 0.73 | 0.003 |
| | 13 | 0.74 | 0.007 |
| | 14 | 0.75 | 0.006 |
| | 15 | 0.76 | 0.005 |
| | 16 | 0.78 | 0.003 |
| | 17 | 0.79 | 0.003 |
| | 18 | 0.80 | 0.003 |
| | 19 | 0.80 | 0.007 |
| | 20 | 0.80 | 0.009 |
| | 21 | 0.80 | 0.011 |
| | 22 | 0.80 | 0.019 |
| | 23 | 0.80 | 0.033 |
| | 24 | 0.83 | 0.006 |
| | 25 | 0.83 | 0.013 |
| | 26 | 0.84 | 0.009 |
| | 27 | 0.85 | 0.014 |
| | 28 | 0.85 | 0.017 |
| | 29 | 0.85 | 0.024 |
| | 30 | 0.85 | 0.035 |

characteristics variate ($C_{M1}$ in Fig 2) with the individual, raw psychological-characteristics measures (individual columns of $C_R$ in Fig 2). We repeated this significance test 23 times, one for each of the 23 first-mode variates defined using the 23 different numbers of PCs (Table 3). Finally, we judged a given measure to be the one that makes a robust contribution to the CCA only when it showed more than 22 significant ($p < 0.05$ with the Benjamini-Hochberg method) correlations with the 23 first-mode variates. As a result, we identified a total of 10 measures. Their across-variate averages of correlations are summarized in Fig 4A. (For the psychological-characteristic measures that failed to meet this rather strict criterion (22 significant results out of 23 tests) but showed at least one significant correlation with the psychological-characteristics variate ($C_{M1}$), see S3A and S3C Fig).

Most of the measures that make robust contributions to the CCA belong to the 'psychosocial factors' class, especially those that are known to reflect the degree to which a given individual receives various kinds of social support from the life environment. The individuals' psychological-characteristics variate ($C_{M1}$) tended to increase as they reported that they receive a wider range of social support (three measures of 'social support scale (SSS)'), are more connected to others ('self-expansion' measure of 'Korean resilience quotient (KRQ)'), are more able to establish and maintain social relationships ('communication' measure of KRQ), have higher degrees of overall self-esteem ('Rosenberg self-esteem scale (RSS)'), experienced more severe and frequent negative life events (two measures of 'life experience survey (LES)') or feel lesser degrees of subjective loneliness and social isolation ('UCLA loneliness scale (ULS)').

Among the measures that do not belong to the 'psychosocial' class, only one measure, 'Audit-K' in the 'clinical-problem' class, robustly contributed to the CCA. The psychological-characteristics variate ($C_{M1}$) tended to increase as the scores of 'Audit-K', which indicates the degree of excessive alcohol drinking, increased.

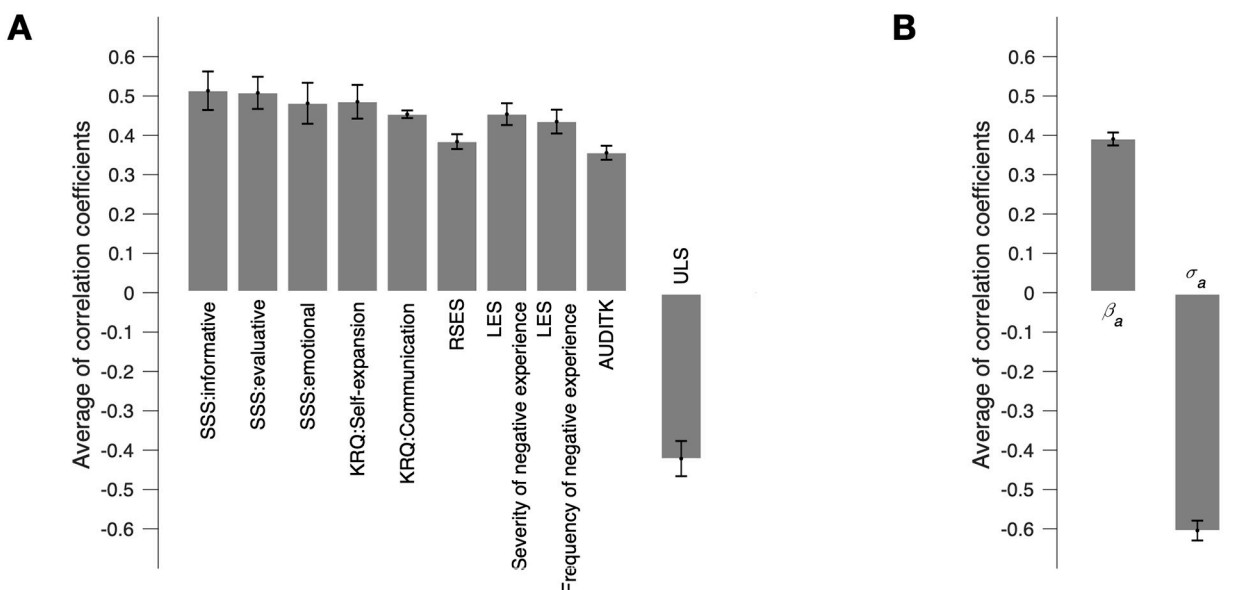

**Fig 4. The contributions of the psychological-characteristics and emotion-response measures to the CCA mode.** (A) The across-variate averages of the correlations between the raw measures of psychological characteristics and the first-mode psychological-characteristics variate ($C_{M1}$). (B) The across-variate averages of the correlations between the raw measures of emotion responses and the first-mode emotion-response variate ($E_{M1}$). (A, B) Only the raw measures that showed significant correlations with more than 22 out of the 23 different CCA variates are shown. Error bars, 95% confidence intervals. SSS, social support scale; KRQ, Korean resilience quotient; RSES, Rosenberg self-esteem scale; LES, life experience survey; Audit-K, Alcohol Use Disorder Identification Test; ULS, UCLA Loneliness Scale.

## The emotion-response measures contributing to the first CCA mode

Using the same procedure and criterion used for the psychological-characteristic measures, we identified the emotion-response measures that make robust contributions to the CCA mode. As a result, two measures in the arousal dimension were identified, the 'accuracy ($\sigma_a$)' and 'sensitivity ($\beta_a$)' measures (Fig 4B). (For the emotion-response measures that showed at least one significant correlation with the emotion-response variate ($C_{M1}$), see S3B and S3D Fig).

The emotion-response variate of the CCA mode ($E_{M1}$) tended to increase as the accuracy measure decreased and the sensitivity measure increased. Since the accuracy measure reflects an extent to which a given individual's responses deviate from the normative responses (the fourth panel in Fig 2I), the negative correlation between $\sigma_a$ and $E_{M1}$ means that the individuals with higher values of the emotion-response variate tended to show the arousal responses that are more deviant from the population-average responses to the visual narrative stimuli. On the other hand, the sensitivity measure reflects an extent to which changes between a given individual's responses to different VN stimuli are greater than those expected from the normative responses to VN stimuli (the second left panel in Fig 2I). Therefore, the positive correlation between $\beta_a$ and $E_{M1}$ means that the individuals with higher values of the emotion-response variate tended to show the arousal responses that are more exaggerated than the population-average responses.

## Polarized arousal responses in the individuals with high CCA variates

Having identified the two arousal measures contributing to the CCA mode, we carried out further analysis to find a critical feature that jointly describes the relationships of the accuracy and sensitivity measures with the CCA mode in a unified manner.

As the first step of the analysis, we classified the individuals into two groups based on their canonical-variate scores ($E_{M1}$) and plotted the group-averaged values of absolute deviation of arousal responses from the normative responses (Fig 5A) and arousal responses (Fig 5B) against the normative responses across the 144 VN stimuli. The canonical-variate scores ($E_{M1}$) used here was the one defined with the CCA based on 27 PCs, which produced the most representative results. The absolute deviations were greater for the almost entire range of the normative responses in the high-variate-score group (Fig 5A) while the reported arousal scores varied more steeply as a function of the normative response in the high-variate-score group (Fig 5B). As a plausible scenario that is coherent with both of these two patterns, we considered the possibility that the individuals with high variate scores tend to show more 'polarized arousal responses' than those with low variate scores. Specifically, the extent to which responses are 'polarized' refers to the extent to which responses are attractively biased toward both of the two extreme poles. Thus, if a given individual's responses are more polarized than the normative responses, her or his responses will be not just more deviant from the normative responses but also more exaggerated than the normative responses.

To confirm the 'polarized-arousal-response' scenario, we compared the distributions of arousal rating scores between the low-variate-score and high-variate-score groups (Fig 5C). As anticipated by the 'polarized-arousal-response' scenario, the response distributions were indeed different between the two groups (see F-test results at the bottom of Fig 5C) and more polarized in the high-variate-score group than in the low-variate-score group (see kurtosis and skewness results at the bottom of Fig 5C). For the merged distributions (the second-rightmost panel of Fig 5C), the kurtosis was significantly lower—i.e., flatter—in the high-variate-score group (1.76) than in the low-variate-score group (2.16). This difference in kurtosis resulted mainly from the fact that the arousal responses were more polarized in the high-variate-score group than in the low-variate-score group. The tendency of making polarized arousal

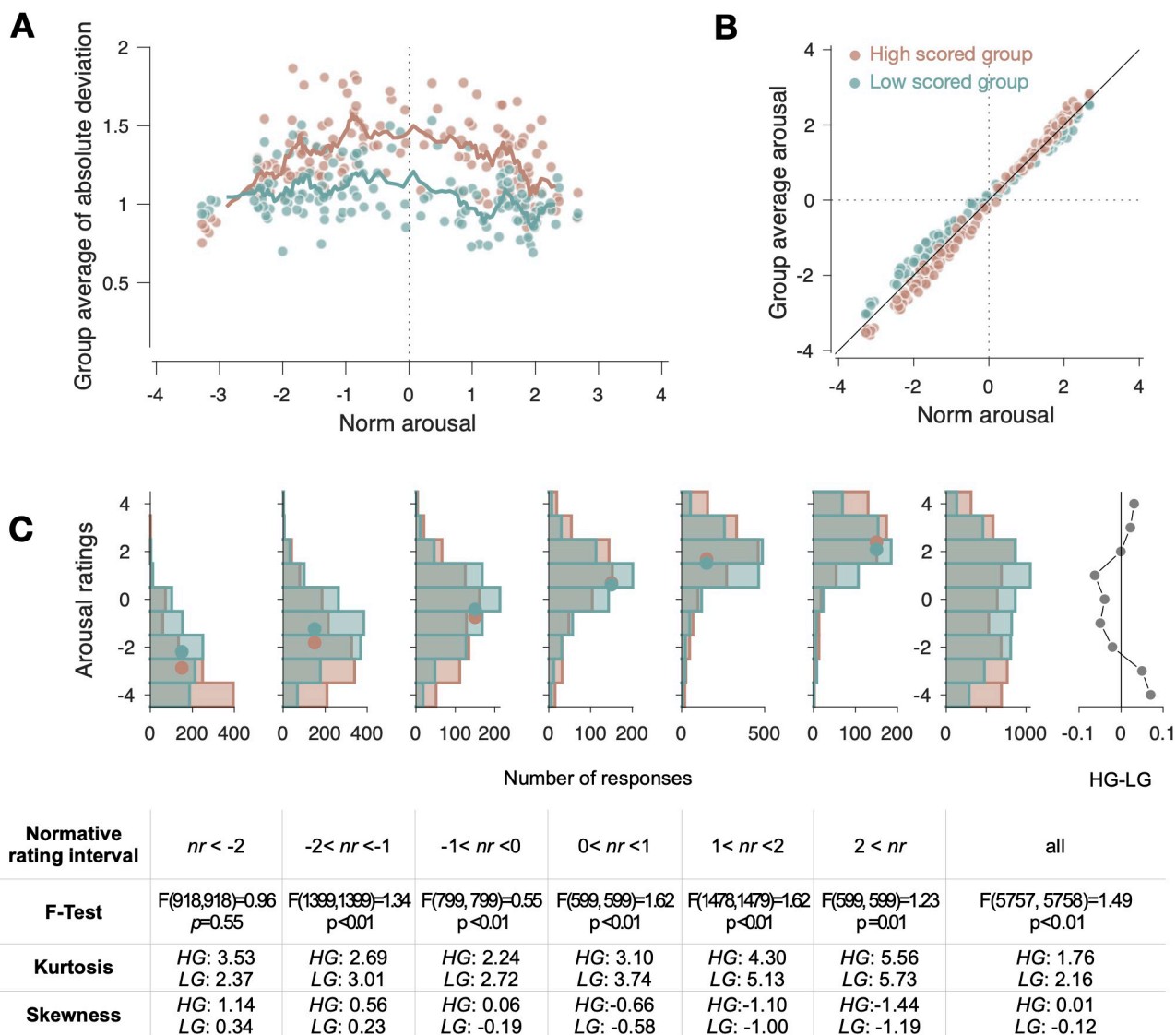

**Fig 5. Polarized arousal responses in the individuals with high CCA variates.** (A) The averaged absolute deviations of arousal responses from the normative responses plotted against the normative responses for the high (maroon dots) and low (teal dots) $E_{M1}$-score groups. The lines are the moving averages (window size, 10) of the averaged absolute deviations. (B) The averaged arousal responses plotted against the normative responses for the high and low $E_{M1}$-score groups. (C) The comparison of the distributions of arousal responses between the high and low $E_{M1}$-score groups. Top, the histograms of arousal responses that are binned according to the normative response (six panels from left), the merged histograms of the entire arousal responses (the second-rightmost panel), and the relative differences in proportion between the merged histograms (the rightmost panel, where 'HG' and 'LG' stand for the high and low $E_{M1}$-score groups, respectively). The histograms for the high and low $E_{M1}$-score groups are shown in maroon and teal, respectively. Bottom, the table summarizes the statistics of the histograms shown above. The columns' locations are matched to the histograms that they describe. $nr$ stands for the normative responses.

responses in the high-variate-score group was also evident in the local distributions that were binned according to the normative responses (the 6 panels from left in Fig 5C). As the range of the normative responses becomes lower or higher (i.e., approaches toward extreme values), the response distributions become more skewed in the high-variate-score group than in the low-variate-score group (as indicated by the skewness values in Fig 5C). On the other hand, as the range of the normative responses approaches toward intermediate values, the response

distributions become flatter in the high-variate-score group than in the low-variate-score group (as indicated by the kurtosis values in Fig 5C).

We also inspected the distributions of the valence responses with the same procedure used for the arousal responses but did not find substantial differences between the high-variate-score group and the low-variate-score group (S4 Fig).

We note that there is a, rather trivial, alternative account for the observed differences in distributions between the high and low $E_{M1}$-score groups: such differences may also arise from the differences in the overall tendency of given individuals to *report* extreme values whatever being measured. To address this issue, we (i) estimated such tendency for the individual participants from their reports in the psychological-characteristics questionnaires, (ii) de-confounded the data for such tendency, and (iii) repeated the CCA analysis (for details, see Methods in S2 Appendix). The results of this de-confounded CCA were quite similar to those of the original CCA: the arousal accuracy ($\hat{p} = 100$, $E(r) = -0.55$) and arousal sensitivity ($\hat{p} = 74\%$, $E(r) = 0.28$) still showed the robust relationship with psychosocial factors similar to our main findings (see S2 Appendix Fig 1 for details). These results suggest that the polarized arousal responses are unlikely to be explained away by the overall tendency of reporting extreme values.

## Comparisons of the pairwise correlations and the results of the multivariate analysis

To directly compare our work with previous work, which mostly took the hypothesis-driven regional approach based on pairwise correlations, and also to further understand the structure of population co-variation between the psychological-characteristics and emotion-response domains, we calculated pairwise Pearson correlations for all the possible pairs between the measures of the two domains (left panel of Fig 6) and compared those correlations with the CCA variates (right panel of Fig 6). By comparing the Pearson correlations and the CCA-variate correlations, all the possible relationships between the measures of the two domains can be classified into four different types, as follows: first, the relationships that were insignificant in both types of correlation; second, those that were significant in Pearson correlation but insignificant in CCA correlation; third, those that were insignificant in Pearson correlation but significant in CCA correlation; lastly, those that were significant in both types of correlation. We note that all the 'significant' Pearson correlations turned out insignificant after being corrected for multiple comparisons (Benjamini-Hochberg correction).

Many (17) relationships fell into the class in which their Pearson correlation was significant but their CCA correlation was insignificant (those marked by empty rectangles in the left panel of Fig 6). On the other hand, four psychological-characteristics measures (KRQ-communication, SSS-informative, ULS, and Audit-K, which are marked by the empty squares in the right panel of Fig 6), despite their significant correlations with the CCA variate, did not show significant Pearson correlations either with the arousal sensitivity measure ($\beta_a$) or with the arousal accuracy measure ($\sigma_a$). On the contrary, six psychological-characteristics measures (KRQ-self-expansion, LES-frequency of negative experience, LES-severity of negative experience, SSS-emotional, SSS-evaluative, and RSES) not just contributed, jointly with the arousal sensitivity ($\beta_a$) and accuracy ($\sigma_a$) measures, to the CCA mode (as marked by the solid squares in the right panel of Fig 6) but also showed significant Pearson correlations with those two emotion-response measures (as marked by the solid rectangles in the left panel of Fig 6). This result, if we put together the signs of Pearson and CCA correlations, helps us interpret a refined structure of the CCA mode. That is, the CCA mode mainly consists of the positive covariation of the arousal sensitivity measure in the emotion-response domain with the KRQ-

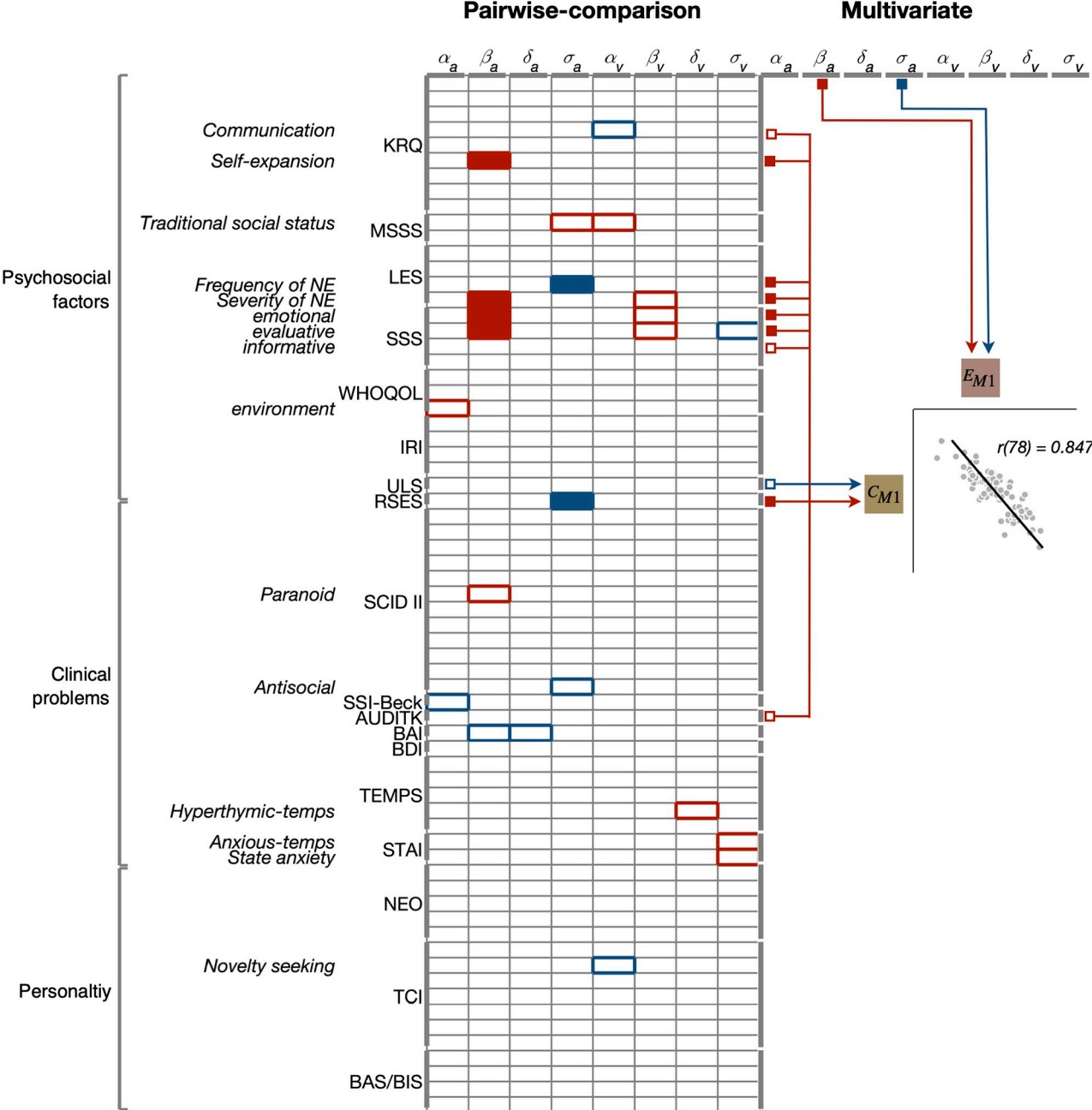

**Fig 6. Comparisons of the pairwise correlations and the results of the multivariate analysis.** Left, the rows and columns of the matrix represent the psychological-characteristics and emotion-response measures, respectively. The empty rectangles mark the pairs of measures that were significant in Pearson correlation but did not make significant contributions to the CCA mode. The solid rectangles mark the pairs of measures that not only were significant in Pearson correlation ($p < 0.05$, uncorrected for multiple comparisons) but also made significant contributions to the CCA mode. Colors of the rectangles indicate the signs of Pearson correlation (red, positive; blue, negative). Right, the schematic structure of the CCA mode illustrated based on the correlations of the measures with the CCA variates. The inset plots the psychological-characteristics variate against the emotion-response variate over individual participants, which is identical to the panel of Fig 2G. The empty squares mark the measures that made significant contributions to the CCA mode but failed to show significant Pearson correlations. The solid squares mark the measures that not only made significant contributions to the CCA mode but also showed significant Pearson correlations. Colors of the squares indicate the signs of correlations with the CCA variates (red, positive; blue, negative). KRQ, Korean resilience quotient; MSSS, MacArthur scale of subjective social status; LES, life experiences survey; SSS, Social Support Scale; WHOQOL, world health organization quality of Life; IRI, interpersonal reactivity index; ULS, UCLA Loneliness Scale; RSES, Rosenberg self-esteem scale; SCID-II, structured clinical interview schedule for DSM-IV Axis-II disorder; SSI-Beck, Beck scale for suicidal ideation; AUDIT-K, Alcohol Use disorder identification test; BAI, Beck anxiety inventory; BDI, Beck depression inventory; TEMPS, temperament evaluation of Memphis, Pisa, Paris and San Diego; STAI, state-trait Anxiety Inventory; NEO, revised NEO personality inventory; TCI, temperament and character inventory; BAS/BAS, behavioral approach/inhibition system.

self-expansion, LES-severity of negative experience, SSS-emotional, and SSS-evaluative in the psychological-characteristics domain (as indicated by the solid red rectangles in the left panel of Fig 6) and the negative covariation of the arousal accuracy measure in the emotion-response domain with the LES-frequency of negative experience, and RSES measures in the psychological-characteristics domain (as indicated by the solid blue rectangles in the left panel of Fig 6).

## Discussion

Being motivated to identify a systematic structure that governs the population covariation between the emotion-response and psychological-characteristics domains, we took a data-driven and global approach by carrying out a series of multivariate analyses on a high-dimensional data set consisting of the eight emotion-response measures and 68 psychological-characteristics measures that were acquired from a cohort of 86 human participants. Having had identified a single, robust, canonical mode of covariation using the CCA in conjunction with PCA, we projected that canonical mode back onto the raw measures in both domains and carried out further analyses to explore 'interpretable' inter-domain relationships underlying the canonical mode. We found one such relationship: individuals who can be characterized by being 'rich in psychosocial assets' tend to show 'polarized arousal responses' to emotion-inducing visual narratives.

### Polarized arousal responses

Emotion differentiation, which is also known as emotion granularity [36], refers to people's ability to distinguish between similar emotions. In the studies which probed categorical emotion responses in the two-dimensional affective space [81, 82], individuals with high emotion differentiability showed emotion responses that were widely distributed mainly along the 'arousal' dimension, which can be interpreted to correspond to the polarized responses in the arousal dimension contributing to the canonical mode in the current work (Fig 5C). On the other hand, another previous work reported that individual differences in emotion differentiation were positively correlated with those who show high degrees of resilience and self-esteem [35, 36], which matches the psychosocial measures contributing to the canonical mode in the current work. Put together, these reports on emotion differentiation appear highly consistent with the canonical mode of population covariation and our interpretation of it.

### Association between psychosocial assets and polarized arousal responses

We conjecture that the observed tight linkage between richness in psychosocial assets and polarized arousal responses might have to do with a phenomenon called "the social sharing of emotions [83]" and an influential view developed upon this phenomenon [84]. According to this view, emotional experiences are not short-lived and intrapersonal but actively shared with other individuals, functioning as social signals of communicating one's internal states, which eventually promotes social interactions. For instance, by crying, a baby can send a parent a signal of hunger, and that signal, in turn, triggers further interactions between the baby and the parent. Supporting this view, intensive emotional experiences are known to be more likely to be expressed to others [85–87] and even discussed with others to some degree [88]. For example, people tend to talk more with strangers after watching together the movies that are emotionally intense than after watching those that are not [89]. According to this view, the individuals who showed more polarized arousal responses to the visual narratives, compared to those who showed less polarized responses, in the current work are more likely to express their emotions to others in their daily life and thus more likely to be engaged in social

interactions. And such increased social interactions would be translated into the high scores on the psychosocial factors that indicate the richness in psychosocial assets, such as those that reflect 'receiving more social supports', 'feeling connected with others', 'having good communication with others', and 'subjective feeling of heightened self-esteem'.

We stress that the proposed account above should be considered as one plausible hypothesis for the observed association between the emotion-response and psychological-characteristics domains. Thus, the validity of this hypothesis must be verified in empirical studies. Especially, it would be ideal if such studies can address the issue of the direction of influences between social interactions and emotion expression, given that our findings do not imply any causal relationship between the emotion-response and psychological-characteristics domains.

## Association between negative life experiences and polarized arousal responses

The CCA mode identified in the current work indicates that the tendency of showing polarized arousal responses was also associated with that of having negative experiences more frequently and severely. As a hypothetical account for this association, we considered a possibility that stressful life events are likely to make individuals react to emotion-inducing stimuli more sensitively. In line with this possibility, it has been reported that reading stressful stories tends to make people better categorize emotions [90, 91], which could be interpreted as increasing the level of attention to emotional events under uncertain and threatening situations [91].

## A negligible contribution of clinical problems to the population covariation

Previous studies reported that some emotion measures are correlated across individuals with the psychological-characteristics measures on mental disorders ('clinical problems' according our labeling scheme), especially the anxiety-related measures [23, 30]. However, the contribution of the clinical-problem measures to the population covariation between the emotion and characteristics domains was almost negligible, if any, in the current work. Although we acquired many ($N$ = 24) clinical-problem measures from a comprehensive set of diverse and representative questionnaires, we found that none of them, except for one ('alcohol-use' measured with Audit-K), significantly contributed to the population covariation. The outcomes of the pairwise comparison analysis (Fig 6, left) suggest one plausible reason for the difference between the previous and current works. Initially, we found many significant pairwise correlations including the clinical-problem measures from the questionnaires such as SCID-II, SSI Beck, BAI, TEMPS, and STAI. However, they all fail to be significant once corrected for multiple comparison. This suggests that the correlations involving clinical-problem measures were reported to be significant in the previous work because they were tested individually in isolation despite not being sufficiently strong to survive the correction for multiple comparisons. To be sure, we do not insist that those pairwise comparisons are inappropriate. They served the main purpose of the previous work, which was to verify specific hypothesis-driven predictions. Our findings suggest that the contribution of clinical-problem measures to the population covariation between the emotion-response and psychological-characteristics domains is not as strong as the psychosocial measures.

As mentioned above, the measure of 'alcohol-use', unlike all the other clinical-problem measures, was significantly correlated with the canonical mode of population covariation. We considered two possible scenarios for this correlation. First, alcohol overuse might have impaired cognitive ability in general, including emotion processing. This scenario seems

consistent with previous clinical studies [92] reporting the correlation between alcohol use disorders and inaccuracy in facial emotion perception, because one feature of polarized emotion responses is the increased deviations from normative (average) response—i.e., inaccuracy in emotion response. As an alternative scenario, it is possible that individuals who are socially active [93] or under stressful situations [94] are prone to alcohol consumption. In line with the latter scenario, the pairwise correlation analysis on our data showed that the alcohol measure was positively correlated both with the frequency of negative experiences ($r = 0.3$, $p < 0.01$) and with the severity of negative experiences ($r = 0.26$, $p = 0.02$).

## No significant relationship of personality measures with emotion measures

Previous studies reported that a few psychological-characteristics measures of personality traits are significantly correlated with emotion responses. For example, it has been reported that valence responses to static images are biased positively and negatively in individuals with extraversion and neuroticism traits, respectively [31, 95]. These previous reports suggest at least some significant pairwise correlations of those trait measures with some of valence measures in our data. However, we could not find such correlations at all, needless to mention no involvement of those traits in the between-domain population covariation. In that regard, none of the remaining measures of personality show significant correlations with any of the emotion response measures or the population covariation either (only one TCI measure (harm avoidance) showed a significant correlation with the bias measure of valence but failed to survive the correction of multiple comparisons). We conjecture that the difference in emotion-inducing stimuli between the previous work (simple static images) and the current work (complex unfolding-over-time narratives) might have resulted in different results. Alternatively, the visual narrative stimuli used in the current work might not have created a sufficient degree of variability in valence responses, as hinted by the standard deviations in the valence measures being somewhat smaller than those in the arousal measures (Table 2).

## Other contributions of the current work to emotion research

Apart from identifying the robust mode of population covariation between the emotion-response and psychological-characteristics domains, the current work makes several useful contributions to the scientific investigation of individual differences in emotion. First, we found that there are substantive individual differences in 'arousal' responses to the same stimuli and that those differences can be predicted by profiling individuals for the psychological-characteristics measures, particularly the psychosocial measures. This warns against the possibility that even the same experimental manipulation of emotion using laboratory stimuli may end up with inducing substantially different degrees of 'subjective (or effective) arousal' responses across individuals. This, in turn, calls for the attention to the necessity of controlling for such individual differences in emotion induction effects by taking into account the tight linkage found in our study between the polarized arousal-response style and the rich profile of psychosocial assets. Second, for the purpose of promoting fine-grained individual differences in emotion responses, we developed a large number (N = 144) of film excerpts that induce a wide range of affective states by visually unfolding stories over time. This library of visual narratives can be used to tap into individual differences in contextual effects on emotion [96, 97] or subtle and nuanced emotion processing, such as emotion granularity [81]. No significant interindividual association between emotion responses and clinical problems was found in our non-patient cohort of subjects. However, such association might be found in clinical populations. In this regard, our library of visual narratives can be considered as a natural—thus

ecologically valid and unobtrusive—means of detecting the emotional symptoms specific to certain psychiatric disorders, such as schizophrenia, which tend to be accompanied by emotion impairment [26, 98]. Lastly, CCA, as one of the popular multivariate analyses, efficiently explores the association between two multivariate collections of variables by finding linear combinations of each collection that maximize a linear correlation coefficient between the two collections. The current work demonstrated the power of CCA in discovering latent structures of covariation hidden in high-dimensional data sets such as psychological-characteristics measures and diverse aspects of emotion responses. Furthermore, as demonstrated previously [78], the current work showed that CCA becomes even more powerful when it is used in conjunction with another multivariate analysis that compresses high dimensional data into a low dimensional space such as PCA, which allowed us to back-project the CCA mode onto raw—thus interpretable—measures.

## Future work

In what follows, we considered a few issues that should be dealt with in future work. Firstly, we considered the possibility that 'polarized arousal responses' might not reflect participants' actual emotion experiences but rather their overall tendency to choose extreme alternatives whatever questionnaire they work on. To address this concern, we conducted control tests by regressing out the "extreme response style" factor and obtained outcomes that are nearly identical to the original results (S5 Fig). Furthermore, the 'extreme response style' hypothesis seems unlikely because if that hypothesis is true, the 'polarized emotion responses' should have been observed not just for the arousal measures but also for the valence measures, which is inconsistent with our findings (S4 Fig). As a future means of ensuring that the 'polarized response style' indeed reflects participants' actual emotion experiences, we consider adding physiological measurements that are tightly linked with 'arousal' states, such as cardiac and skin-conductance responses. Secondly, we considered how generalizable our finding is to other populations, different cultures in particular. We intentionally recruited participants to form a culturally homogenous population to minimize the individual differences in emotion responses due to cultural differences. It would be important to check whether our findings can be replicated especially in different-age groups or non-Asian cultures. If replicated, the canonical mode of population covariation found in the current work can be considered a highly generic feature of the relationship between psychological characteristics and emotion responses. Even if not replicated, such differences between cultures will provide us with valuable information about cultural differences in emotion processing. Thirdly, we considered possible ways of extending the methods of the current work to translational research on emotion in clinical populations such as schizophrenia or mood and anxiety disorder patients. For example, given the suggested linkage between emotion recognition and social functioning in schizophrenia patients [99] or between negative valence bias and maladaptive social functioning in mood and anxiety disorder patients [100, 101], the CCA analysis may help reveal a robust and refined covariation structure relating certain aspects of emotion responses to the disorder subtypes, symptoms, spectra, or stages. In addition, given the suggested linkage between schizophrenia and difficulty in integrating emotion perception with context [102], the VN stimuli used in the current work seem more suitable for further specifying the exact nature of emotional deficit in people with schizophrenia. Lastly, we considered extending the current work to include brain-imaging measures such as anatomical structure or functional connectivity as a third domain of the multi-domain CCA. Such extensions will help us elucidate the neural basis for the canonical mode of population covariation between the emotion-response and psychological characteristic domains.

## Supporting information

**S1 Fig. Reliability of emotion measures over visual narrative stimuli.**
(DOCX)

**S2 Fig. Invariance of emotion measures to different ways of defining normative emotion responses.**
(DOCX)

**S3 Fig. The detailed contributions of the psychological characteristics and emotion-response measures to the CCA mode.**
(DOCX)

**S4 Fig. Distribution analysis on the valence responses.**
(DOCX)

**S5 Fig. The results of the CCA analysis in which 'extreme response style' was regressed out.**
(DOCX)

**S1 Table. Specifications of psychological-characteristic measures: Psychosocial factors category.**
(DOCX)

**S2 Table. Specifications of psychological-characteristic measures: Clinical problems category.**
(DOCX)

**S3 Table. Specifications of psychological-characteristic measures: Personality category.**
(DOCX)

**S1 File. List of visual narratives.**
(XLSX)

**S1 Appendix. Power analysis and sample size justification.**
(DOCX)

**S2 Appendix. De-confounding the CCA for the overall tendency of making extreme reports.**
(DOCX)

## Author Contributions

**Conceptualization:** Jinyoung Kim, Ji-Won Hur, Jun Soo Kwon, Sang-Hun Lee.

**Data curation:** Jinyoung Kim.

**Formal analysis:** Jinyoung Kim, Eunseong Bae.

**Funding acquisition:** Jinyoung Kim, Sang-Hun Lee.

**Investigation:** Jinyoung Kim, Eunseong Bae, Sang-Hun Lee.

**Methodology:** Jinyoung Kim, Eunseong Bae, Yeonhwa Kim, Chae Young Lim, Sang-Hun Lee.

**Project administration:** Jinyoung Kim.

**Resources:** Jinyoung Kim, Yeonhwa Kim, Jun Soo Kwon.

**Software:** Jinyoung Kim.

**Supervision:** Jinyoung Kim, Sang-Hun Lee.

**Validation:** Jinyoung Kim, Chae Young Lim, Ji-Won Hur.

**Visualization:** Jinyoung Kim.

**Writing – original draft:** Jinyoung Kim, Sang-Hun Lee.

**Writing – review & editing:** Jinyoung Kim, Sang-Hun Lee.

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
