## [Decision Letter · Decision Letter 0]

23 Sep 2021

PONE-D-21-14011A robust multivariate structure of interindividual covariation between psychosocial characteristics and arousal responses to visual narrativesPLOS ONE

Dear Dr. Lee,

Thank you for submitting your manuscript to PLOS ONE. After careful consideration, we feel that it has merit but does not fully meet PLOS ONE’s publication criteria as it currently stands. Therefore, we invite you to submit a revised version of the manuscript that addresses the points raised during the review process.

We look forward to receiving your revised manuscript.

Kind regards,

Christos Papadelis, Ph.D.

Academic Editor

PLOS ONE

Journal Requirements:

2. We noted in your submission details that a portion of your manuscript may have been presented or published elsewhere. "Bae et al., 2020" Please clarify whether this publication was peer-reviewed and formally published. If this work was previously peer-reviewed and published, in the cover letter please provide the reason that this work does not constitute dual publication and should be included in the current manuscript.

Reviewers' comments:

Reviewer's Responses to Questions

**Comments to the Author**

1. Is the manuscript technically sound, and do the data support the conclusions?

Reviewer #1: Partly

Reviewer #2: Yes

2. Has the statistical analysis been performed appropriately and rigorously? 

Reviewer #1: Yes

Reviewer #2: Yes

3. Have the authors made all data underlying the findings in their manuscript fully available?

Reviewer #1: Yes

Reviewer #2: Yes

4. Is the manuscript presented in an intelligible fashion and written in standard English?

Reviewer #1: No

Reviewer #2: Yes

5. Review Comments to the Author

Reviewer #1: The article proposes a method to analyze individual differences in emotion response and explore the systematic structures underlying such response. The authors introduce justification of the existence of such differences or variation and explain the parameters of their analysis and the limitations of previous analyses. In turn, they propose a data-driven approach based on multivariate analyses allowing to identify significant across-individual covariations between the domain of emotion response and that of psychological characteristics. However, in its current form, the paper still has some weaknesses that should be addressed before publication. Firstly, the English needs to be revised to improve intelligibility and eliminate some errors mostly found in the second part of the paper as compared with the first part (see below for specific examples). I would also recommend the authors to avoid criticisms that may sound like overgeneralizations and simplifications (see below for specific examples). One of their major claims is that their method is more powerful than previous analyses based on pairwise comparisons. Everyone who has worked with pairwise comparisons is probably aware of the fact that they can vary depending on the number of factors analyzed. However, even if the multivariate method proposed is certainly more powerful, results also depend on the number of factors introduced. There is, therefore, no need to be too negative about previous approaches; the authors could still emphasize the advantages of their method and recognize the value of previous ones together with their shortcomings. Secondly, even if research meets all applicable standards for the ethics of experimentation and the statistical analysis seems to be described in sufficient detail, there are still details from the experiment that should be provided to clarify the protocol, especially those referring to the criteria to select stimuli and materials (specific examples are also provided below). Thirdly –and this is actually one of my major concerns– the discussion of results somehow lacks rigor and theoretical justification. Even if I understand that the data-driven approach adopted aims at avoiding theoretical bias, plausible explanations for the findings should be soundly justified and grounded in the literature; or alternatively, they could be provided just as hypotheses to be verified in future studies. Finally, despite its interest and value, the method proposed seems too complex and time consuming when comparing effort vs results. The benefits of using the method could be further outlined to make it worthwhile.

Here is a list of some of the changes required at specific points in the paper:

LANGUAGE AND STYLE

-Lines 105-108. The first part of the sentence requires a verb: “The variability of emotion refers to how inconsistent an individual’s emotion responses to the same event or highly similar events and…”

-Lines 118-120. I would recommend the authors to modulate and soften the criticism against “all the previous studies”, unless they positively know that they all tested “predictions of interest”. Their assumption that testing particular theories or hypotheses is necessarily negative because it leads authors to focus on “a few pre-selected measures” (lines 150-152) seems an oversimplification that may not be always true. I would also extend my disagreement to their assumption that all previous studies use pairwise correlations between single emotion measures and single psychological characteristics measures (lines 162-165). I think that the authors could make their point and avoid this type of overstatements.

-Lines 505 and 600. The expression “As results” is not clear in this context. Please, rephrase to “As a result” or a near synonym.

-Line 539. The correct form of the verb “remain” in “only the first CCA mode remain significant” should be either “remains” in the present or “remained” in the past.

-Line 586. A comma is missing before “have higher degrees of overall self-esteem”

-Lines 635-637. The sentence “we considered the possibility that the individuals with high variate scores tend to show ‘polarized arousal responses’ than those with low variate scores do” is missing a “more” or a “less”. Besides, the “do” at the end is not necessary

-Lines 639-641. In the sentence “Thus, if a given individual’s responses are more polarized than the normative responses, her or his responses will be not just more deviant from the normative responses but also be more exaggerated than the normative responses” the “be” in the second part can be omitted.

-Lines 668-669. The verb “is” should be deleted from the sentence: “This difference in kurtosis is resulted mainly”

-Line 793. “with those who shows” should be replaced with “with those who show”

-Lines 817-819. I would recommend to rephrase the whole sentence to enhance comprehension

-Lines 828-829. I would also recommend to rephrase the sentence “were found to make people better categorize emotions”

-Lines 845-847. The whole sentence needs to be rephrased, since either a whole clause seems to be missing or “that” should be deleted: “Initially, we found that many significant pairwise correlations including the clinical-problem measures from the questionnaires such as SCID-II, SSI Beck, BAI, TEMPS, and STAI.”

-Lines 851-857. The English should be revised here and in the whole section: e.g. “our findings indicate that the clinical-problem measures do not as strongly participate as the psychosocial measures in the canonical mode that governs the population covariation”

-Lines 916-919. The sentence needs to be rephrased, since the last part does not make any sense. It probably needs a “which” after “disorders”: “our library of visual narratives can be considered as a natural—thus ecologically valid and unobtrusive—means of detecting the emotional symptoms specific to certain psychiatric disorders, such as schizophrenia, tend to be accompanied by emotion impairment”. Besides, no sound arguments are provided for their proposal to use their library of visual narratives to detect emotional symptoms specific to psychiatric disorders, especially considering the lack of significant results provided in their study for these variables.

-I would recommend the authors to avoid the use of expressions such as “a previous study” when referring to previous results. Without introducing the specific work, the sentence sounds somewhat clumsy. The number reference to the work in the reference list can be inserted without resorting to these expressions.

-Lines 919-923. I would recommend the authors to avoid advising against the use of the type of methods they are advocating for in the present study, at least in the way they have phrased the warning: “despite the power of discovering latent structures of covariation hidden in high dimensional data sets, multivariate analysis methods have not to be exercised frequently due to the curse-of-dimensionality problem and the difficulty in interpretation”.

-Lines 929-930. I would delete “before concluding the current work” from the sentence since it impairs comprehension.

-Line 947. I would avoid referring to another age group as a different culture.

INFORMATION

-Lines 126-130. Authors could maybe provide specific references to some meta-analyses of the kind they point to.

-Lines 230-235. Further details about the protocol would be appreciated. Did participants take the 19 questionnaires home to complete them without any specific instructions on how to do it? Were they advised to fill them in at different times to avoid exhaustion? Also, the criteria to select them seem rather loose, since the explanation provided only specifies that they “were developed with different taxonomies that capture individual differences in relatively enduring behavioral tendencies from diverse theoretical and practical perspectives” (231-33).

-In relation to the questionnaires used, no information is provided as to whether the authors used the Korean version or the English one. Reference to the use of the Korean version is found in S1_Table (although not for every questionnaire), but it should also be clarified in the text.

-Lines 262-263. I cannot see the logic of including 4 music videos and 4 TV commercials in the corpus, since the numbers are hugely unbalanced in comparison with the 116 motion pictures. This unbalance should be further justified.

-Lines 271-273. I also have problems with the relevance of selecting a “coherent piece of storytelling, so that it could be readily narrated with a few sentences”, considering that all stimuli were made soundless (line 276).

-Lines 302-303. The authors affirm that they confirmed that the “stimuli covered the affective space in a representative manner”. However, no data are provided of how they did so.

JUSTIFICATION

Lines 783-831. I appreciate and value the authors’ effort to provide explanations for the link between polarized arousal responses and psychosocial factors, but their arguments lack strength and theoretical support. They refer to previous studies in the literature and also to a particular theoretical view, but the justification and links should be strengthened. For example, on lines 798-800 the authors state to consider “one influential view” based on the impact of emotion on social interaction “as a possible explanation for the tight linkage between psychosocial factors and polarized arousal responses”. However, on lines 817-819 they acknowledge that an explanation in the opposite direction is also possible: “allowing for an interpretation based on the impact of social interactions on emotion expression in the opposite direction of influence posited by the view introduced above”. Furthermore, on line 823 they mention another result that cannot be easily interpretable based on these explanations and briefly mention another line of previous work. Further than the plausibility of the explanations provided, the final impression is that possibilities are suggested as they are found, but no theoretical background or standpoint is used to justify them.

Reviewer #2: The author's work is interesting and methodologically well-done. The investigation of individual covariations between psychological and emotional (arousal and valence) constructs in visual narratives is fascinating, and a needed step in the field of human function dimensions. Moreover, this understanding can be applied to many populations, such as mood and anxiety disorders. While the manuscript is intelligible, my main comments are larger, broad suggestions to improve the formatting (some focus) and readability of the manuscript.

Below I have outlined points that would improve the manuscript:

1) The abstract needs to be rewritten and model the formatting of typical abstracts. It lacks details from the methods (participant information) and more specific details of the results, and their interpretation. The abstract is purpose or "introduction" heavy. I finished reading it unsure of what the project actually did and found.

2) The manuscript is generally well written. My concern is that the manuscript reads, and is formatted, closer to that of a thesis or dissertation paper, rather than a targeted journal article (e.g., how the introduction starts; many details in the methods could be supplemental – like power and sample size).

3) The Figure captions being included within the body of the manuscript might be allowed for the journal submission, but they would be better suited in a "Figure Caption" section if able.

4) As for the figures, reconsider what items are necessary and those that are not (e.g., Fig 1 – A, B, C, D and F do not seem necessary to me; Fig 5 - skewness, kurtosis etc is not needed here). Demographics can be tabled, and would be easier to understand. There should be more room dedicated to visualizing your data/results.

5) One quick statement in the discussion about how the work could be utilized in populations like schizophrenia opens the door for a very important application that the current work should highlight more. Performing this work in mood and anxiety disorders is a natural next step that is not discussed in the future work section at the end of the discussion. It should be, and projected findings even suggested.

6. PLOS authors have the option to publish the peer review history of their article (what does this mean?). If published, this will include your full peer review and any attached files.

Reviewer #1: **Yes: **Ana María Rojo López

Reviewer #2: No

---

## [Author Response · Author response to Decision Letter 0]

10 Nov 2021

We thank both reviewers for the highly constructive comments, which provided us with an opportunity to improve our manuscript's intelligibility and readability. In below, we briefly summarize the gist of our revision.

Reviewer #1 systematically sorted a total of 26 comments into four different issues, “language and style (18 comments)”, “information (6 comments)”, “justification (1 comment)”, and “benefits of the multivariate approach (1 comment)”. Since we found that all the comments were written clearly and had relevant points, we have incorporated all of them into the revised manuscript. To list major revision points, we (i) substantially revised INTRODUCTION to avoid “oversimplification”, “overgeneralization”, and unnecessary negative attitude toward previous studies as for “language and style”; (ii) added the further details of our protocol, such that our study can be replicated based on our description, as for “information”; (iii) substantially revised DISCUSSION to indicate explicitly and exactly what aspects of findings might be explained by which theoretical viewpoints respectively, as for “justification”; (iv) further specified the benefits of the multivariate approach based on what was demonstrated in the current work, as for the last comment.

Reviewer #2’s comments (a total of 5 comments) were mostly about the format and readability of the manuscript. We agree with all these comments and incorporated them into the revised manuscript except for one regarding the location of figure captions (we could not do that due to the PLOS ONE format guideline). Consequently, Abstract and Fig 1 were substantially revised, INTRODUCTION and DISCUSSION were modified, and two Methods sections were now presented as Supplementary Appendices.

For an effective presentation and cross-referencing of our replies to the two reviewers’ comments, we labeled each of them with a format [C#-#-#], where #s represent the identity of a reviewer, the category of the comment, and the order of the comment within that category. To distinguish between the comments and our replies, italic fonts were used for the comments.

Replies to Reviewer #1’s comments

[Overall comment]: “The article proposes a method to analyze individual differences in emotion response and explore the systematic structures underlying such response. The authors introduce justification of the existence of such differences or variation and explain the parameters of their analysis and the limitations of previous analyses. In turn, they propose a data-driven approach based on multivariate analyses allowing to identify significant across-individual covariations between the domain of emotion response and that of psychological characteristics. However, in its current form, the paper still has some weaknesses that should be addressed before publication.”

Our reply to [Overall comment]: We thank Reviewer #1 for correctly and succinctly summarizing the core of our work presented in the submitted manuscript. We agree with Reviewer #1 that the previous version of the manuscript “has some weaknesses that should be addressed” and did our best to address those weaknesses in the revised manuscript.

[C1-1] General comment on the issue of “language and style”: “Firstly, the English needs to be revised to improve intelligibility and eliminate some errors mostly found in the second part of the paper as compared with the first part (see below for specific examples). I would also recommend the authors to avoid criticisms that may sound like overgeneralizations and simplifications (see below for specific examples). One of their major claims is that their method is more powerful than previous analyses based on pairwise comparisons. Everyone who has worked with pairwise comparisons is probably aware of the fact that they can vary depending on the number of factors analyzed. However, even if the multivariate method proposed is certainly more powerful, results also depend on the number of factors introduced. There is, therefore, no need to be too negative about previous approaches; the authors could still emphasize the advantages of their method and recognize the value of previous ones together with their shortcomings.” 

Our reply to [C1-1]: We admit that there were many errors in the previous manuscript. In the revised manuscript, we got rid of those grammatical errors and rewrote several sentences, including those pointed out by the reviewer, to improve readability. We greatly appreciate the reviewer’s time and effort to detect specific errors and make suggestions for improvement. Moreover, we also agree with the reviewer that we were “too negative about previous approaches” in the previous manuscript. We rewrote an entire paragraph of Introduction and several other sentences or phrases throughout the manuscript to incorporate the reviewer’s suggestion for addressing this problem (“… emphasize the advantages of their method and recognize the value of previous ones together with their shortcomings.”). For details, please see below for our replies to the reviewer’s specific comments and the corresponding revisions. 

[C1-1-1~18] Specific comments on the issue of “language and style”

[C1-1-1]: Lines 105-108. The first part of the sentence requires a verb: “The variability of emotion refers to how inconsistent an individual’s emotion responses to the same event or highly similar events and…”

Reply to [C1-1-1]: We revised the sentence accordingly (see lines 99-102 in the revised manuscript).

[C1-1-2]: Lines 118-120. I would recommend the authors to modulate and soften the criticism against “all the previous studies”, unless they positively know that they all tested “predictions of interest”. Their assumption that testing particular theories or hypotheses is necessarily negative because it leads authors to focus on “a few pre-selected measures” (lines 150-152) seems an oversimplification that may not be always true. I would also extend my disagreement to their assumption that all previous studies use pairwise correlations between single emotion measures and single psychological characteristics measures (lines 162-165). I think that the authors could make their point and avoid this type of overstatements.

Reply to [C1-1-2]: Although we did not intend to claim that “all the previous studies (that have studied the relationship between the two domains)” tested only “predictions of interest” in the previously submitted manuscript, we admit that the way we contrasted the “hypothesis-driven and regional” approach and the “data-driven and global” approach gave an unnecessary impression of “oversimplification” and “overstatements.” To address this problem, we substantially revised the 5th and 6th paragraphs of Introduction by precisely specifying what previous studies were referred to whenever we refer to “previous studies” (e.g., lines 107, 113) and by avoiding any unnecessary (oversimplified or overstated) contrast with “previous studies” (lines 107-130). 

[C1-1-3]: 1-3 Lines 505 and 600. The expression “As results” is not clear in this context. Please, rephrase to “As a result” or a near synonym.

Reply to [C1-1-3]: We searched the entire manuscript for the wrong use of “as results”, including those detected by the reviewers, and replaced with “as a result” (lines 296, 377,

436, 531).

[C1-1-4]: Line 539. The correct form of the verb “remain” in “only the first CCA mode remain significant” should be either “remains” in the present or “remained” in the past.

Reply to [C1-1-4]: “remain” has been replaced with “remains” (line 470).

[C1-1-5]: Line 586. A comma is missing before “have higher degrees of overall self-esteem”.

Reply to [C1-1-5]: A comma has been added (lines 517-518).

[C1-1-6]: Lines 635-637. The sentence “we considered the possibility that the individuals with high variate scores tend to show ‘polarized arousal responses’ than those with low variate scores do” is missing a “more” or a “less”. Besides, the “do” at the end is not necessary

Reply to [C1-1-6]: The whole sentence has been corrected for those errors (lines 566-568). 

[C1-1-7]: Lines 639-641. In the sentence “Thus, if a given individual’s responses are more polarized than the normative responses, her or his responses will be not just more deviant from the normative responses but also be more exaggerated than the normative responses” the “be” in the second part can be omitted.

Reply to [C1-1-7]: “be” has been deleted (lines 570-572).

[C1-1-8]: Lines 668-669. The verb “is” should be deleted from the sentence: “This difference in kurtosis is resulted mainly”

Reply to [C1-1-8]:]: “is” has been deleted (lines 599-601).

[C1-1-9]: Line 793. “with those who shows” should be replaced with “with those who show”

Reply to [C1-1-9]: “shows” has been replaced with “show” (line 722).

[C1-1-10]: Lines 817-819. I would recommend to rephrase the whole sentence to enhance comprehension

Reply to [C1-1-10]: This phrase (“… thus allowing for an interpretation based on the impact of social interactions on emotion expression in the opposite direction of influence posited by the view introduced above …”) has been deleted to address one of the reviewer’s other comments regarding the “justification” issue. For the details of revision, see our Reply to [C1-3] below.

[C1-1-11]: Lines 828-829. I would also recommend to rephrase the sentence “were found to make people better categorize emotions”

Reply to [C1-1-11]: The sentence has been rephrased as follows: “In line with this possibility, it has been reported that reading stressful stories tends to make people better categorize emotions” (lines 765-767).

[C1-1-12]: Lines 845-847. The whole sentence needs to be rephrased, since either a whole clause seems to be missing or “that” should be deleted: “Initially, we found that many significant pairwise correlations including the clinical-problem measures from the questionnaires such as SCID-II, SSI Beck, BAI, TEMPS, and STAI.”

Reply to [C1-1-12]: “that” has been deleted (lines 783-785).

[C1-1-13]: Lines 851-857. The English should be revised here and in the whole section: e.g. “our findings indicate that the clinical-problem measures do not as strongly participate as the psychosocial measures in the canonical mode that governs the population covariation”

Reply to [C1-1-13]: The last three sentences of this section have been revised, as follows: “To be sure, we do not insist that those pairwise comparisons are inappropriate. They served the main purpose of the previous work, which was to verify specific hypothesis-driven predictions. Our findings suggest that the contribution of clinical-problem measures to the population covariation between the emotion-response and psychological-characteristics domains is not as strong as the psychosocial measures.” (lines 789-794)

[C1-1-14]: Lines 916-919. The sentence needs to be rephrased, since the last part does not make any sense. It probably needs a “which” after “disorders”: “our library of visual narratives can be considered as a natural—thus ecologically valid and unobtrusive—means of detecting the emotional symptoms specific to certain psychiatric disorders, such as schizophrenia, tend to be accompanied by emotion impairment”. Besides, no sound arguments are provided for their proposal to use their library of visual narratives to detect emotional symptoms specific to psychiatric disorders, especially considering the lack of significant results provided in their study for these variables.

Reply to [C1-1-14]: We understood the reviewer’s point and admitted that the sentences in the previous manuscript sounded conflicting with no significant contribution of clinical-problem measures to the CCA mode (, which was summarized in the Discussion section titled “A negligible contribution of clinical problems to the population covariation”). However, this result was obtained from non-patient people, and our VN stimuli can still be considered in translational research on emotion deficit in people with mental diseases. In the revised manuscript, we explicated pointed out this possibility (lines 851-857). 

[C1-1-15]: I would recommend the authors to avoid the use of expressions such as “a previous study” when referring to previous results. Without introducing the specific work, the sentence sounds somewhat clumsy. The number reference to the work in the reference list can be inserted without resorting to these expressions. 

Reply to [C1-1-15]: We agree with and appreciate the reviewer’s recommendation. Accordingly, we searched the entire manuscript for such “clumsy” uses of “a previous study” and revised them as recommended. 

[C1-1-16]: Lines 919-923. I would recommend the authors to avoid advising against the use of the type of methods they are advocating for in the present study, at least in the way they have phrased the warning: “despite the power of discovering latent structures of covariation hidden in high dimensional data sets, multivariate analysis methods have not to be exercised frequently due to the curse-of-dimensionality problem and the difficulty in interpretation”.

Reply to [C1-1-16]: We agree with and appreciate the reviewer’s recommendation. Accordingly, the sentence was revised as follows: “… the current work showed that CCA becomes even more powerful when it is used in conjunction with another multivariate analysis that compresses high dimensional data into a low dimensional space such as PCA, which allowed us to back-project …” (lines 864-868).

[C1-1-17]: Lines 929-930. I would delete “before concluding the current work” from the sentence since it impairs comprehension.

Reply to [C1-1-17]: “before concluding the current work” has been deleted (lines 872-873). 

[C1-1-18]: Line 947. I would avoid referring to another age group as a different culture.

Reply to [C1-1-18]: We agree with and appreciate the reviewer’s recommendation. Accordingly, the sentence has been revised as recommended (lines 888-889)

[C1-2] General comment on the issue of “information”: “ Secondly, even if research meets all applicable standards for the ethics of experimentation and the statistical analysis seems to be described in sufficient detail, there are still details from the experiment that should be provided to clarify the protocol, especially those referring to the criteria to select stimuli and materials (specific examples are also provided below).”

Reply to [C1-2]: Thanks to the reviewer’s comments, we now realize that the previous manuscript did not provide sufficient specific details regarding the experimental protocol, including those pointed out by the reviewer. In the revised manuscript, we added further details of our protocol such that our study can be replicated based on our description. For details, see our replies to the reviewer’s specific comments below.

[C1-2-1~6] Specific comments on the issue of “information”

[C1-2-1]: Lines 126-130. Authors could maybe provide specific references to some meta-analyses of the kind they point to.

Reply to [C1-2-1]: To address one of the reviewer’s comments on the issue of language and style ([C1-1-2]), we substantially revised the paragraphs in which the sentences dealing with “meta-analyses” were originally included (see Reply to [C1-1-2] for details). In doing so, we decided not to bring up the issue related to “meta-analyses.”

[C1-2-2]: Lines 230-235. Further details about the protocol would be appreciated. Did participants take the 19 questionnaires home to complete them without any specific instructions on how to do it? Were they advised to fill them in at different times to avoid exhaustion? Also, the criteria to select them seem rather loose, since the explanation provided only specifies that they “were developed with different taxonomies that capture individual differences in relatively enduring behavioral tendencies from diverse theoretical and practical perspectives” (231-33).

Reply to [C1-2-2]: As requested by the reviewer, we provided further details about the protocol of acquiring the psychological-characteristics measures (lines 182-206). Specifically, we detailed how participants were instructed to complete the questionnaires (lines 200-206) and how we selected the questionnaires by specifying the types and contents of the three taxonomies used in the current work (lines 185-196). 

[C1-2-3]: In relation to the questionnaires used, no information is provided as to whether the authors used the Korean version or the English one. Reference to the use of the Korean version is found in S1_Table (although not for every questionnaire), but it should also be clarified in the text.

Reply to [C1-2-3]: As recommended by the reviewer, we’ve explicitly stated that the Korean version of the questionnaires was used (lines 198-199). 

[C1-2-4]: Lines 262-263. I cannot see the logic of including 4 music videos and 4 TV commercials in the corpus, since the numbers are hugely unbalanced in comparison with the 116 motion pictures. This unbalance should be further justified.

Reply to [C1-2-4]: In the revised manuscript, we first further specified the numbers of VN stimuli for each type of source (130 clips from 124 different motion pictures, 7 clips from 4 different music videos, and 7 clips from 4 different TV commercials; lines, 221-225), explained why we ended up excerpting 14 clips by referring to these non-motion-picture sources (lines 222-225), and justified why we think it does not matter to include those clips for the purpose of the current work (lines 225-228), and carried out additional analysis to show that the emotion-response measures remained almost identical regardless of whether those non-motion-picture clips were included or not in the VN library (lines, 228-230; S2 Fig in the revised manuscript).

[C1-2-5]: Lines 271-273. I also have problems with the relevance of selecting a “coherent piece of storytelling, so that it could be readily narrated with a few sentences”, considering that all stimuli were made soundless (line 276).

Reply to [C1-2-5]: We regret that we used the phrase “it could be readily narrated” in the previous manuscript. Our intention was to specify one of the criteria for selecting clips for the VN stimuli, which was that each clip describes a story that can be understood without sound information. In the revised manuscript, we replaced “it could be readily narrated” with “could be readily described” and added a sentence describing our intention to use this criterion (lines 216-218).

[C1-2-6]: Lines 302-303. The authors affirm that they confirmed that the “stimuli covered the affective space in a representative manner”. However, no data are provided of how they did so.

Reply to [C1-2-6]: As recommended, we provided the ground for the statement (“stimuli covered the affective space in a representative manner”) in the revised manuscript. Specifically, we referred to two previous studies which show how affective states are typically distributed in the affective space and indicated that a similar pattern of distribution was observed in the current data by referring to Fig 1A, B in the revised manuscript (lines 250-254).

[C1-3] Comment on the issue of “justification”: “Thirdly –and this is actually one of my major concerns– the discussion of results somehow lacks rigor and theoretical justification. Even if I understand that the data-driven approach adopted aims at avoiding theoretical bias, plausible explanations for the findings should be soundly justified and grounded in the literature; or alternatively, they could be provided just as hypotheses to be verified in future studies.”

“Lines 783-831. I appreciate and value the authors’ effort to provide explanations for the link between polarized arousal responses and psychosocial factors, but their arguments lack strength and theoretical support. They refer to previous studies in the literature and also to a particular theoretical view, but the justification and links should be strengthened. For example, on lines 798-800 the authors state to consider “one influential view” based on the impact of emotion on social interaction “as a possible explanation for the tight linkage between psychosocial factors and polarized arousal responses”. However, on lines 817-819 they acknowledge that an explanation in the opposite direction is also possible: “allowing for an interpretation based on the impact of social interactions on emotion expression in the opposite direction of influence posited by the view introduced above”. Furthermore, on line 823 they mention another result that cannot be easily interpretable based on these explanations and briefly mention another line of previous work. Further than the plausibility of the explanations provided, the final impression is that possibilities are suggested as they are found, but no theoretical background or standpoint is used to justify them.”

Reply to [C1-3]: We thank the reviewer for appreciating our effort to offer some explanations for the results although the current work, as a data-driven study, was not designed to test specific theories or hypotheses. At the same time, we concur with the reviewer that we still need to offer a set of coherent explanations or, at least, do not seem to conflict with one another. We do not think that any single theoretical view can offer a unified account for the CCA mode observed in the current work. We believe that the across-individual covariation structure governing the relationship between the emotion-response and psychological-characteristics domains is likely to be complex in nature and entails diverse, multiple factors. Guided by this belief, we took the strategy of offering a primary explanation (the account based on “social sharing of emotions”) for the most prominent relationship (covariation structure between polarized arousal responses and psychosocial-asset groups of measures) and then offering a secondary explanation (the account based on “stressful life events”) for the (minor) relationship (covariation structure between polarized arousal responses and the negative-experience measures). We do not think that these two explanations conflict with each other because they account for two different aspects of the CCA mode found in the current work.

 Having clarified our perspective on this issue, we regret that three different aspects of our findings (and possible accounts for those aspects) were provided under a single section titled - rather biasedly - “Association between psychosocial factors and polarized arousal response” in the previously submitted version of the manuscript. This contributed to the impression that several incoherent explanations are offered to one single aspect of the findings. Thus, to address this problem in our revised manuscript, we dissected the original section into three sections with respective titles indicating explicitly and exactly what aspects of findings are addressed and offered the corresponding accounts, respectively (lines 713-769). 

[C1-4]: Comment on “benefits of using the multivariate approach”

“Finally, despite its interest and value, the method proposed seems too complex and time consuming when comparing effort vs results. The benefits of using the method could be further outlined to make it worthwhile.”

Reply to [C1-4]: We agree with and appreciate the reviewer’s suggestion. Accordingly, we further specified the benefits of the multivariate approach based on what was demonstrated in the current work (lines 857-868). (By the way, although our method (CCA+PCA) may look complicated, it can be simply considered as an extension of the Pearson correlation analysis. In addition, we believe one can readily apply our method to multivariate datasets using our MATLAB codes available in the OSF repository.)

Replies to Reviewer #2’s comments

[Overall comment]:“The author's work is interesting and methodologically well-done. The investigation of individual covariations between psychological and emotional (arousal and valence) constructs in visual narratives is fascinating, and a needed step in the field of human function dimensions. Moreover, this understanding can be applied to many populations, such as mood and anxiety disorders. While the manuscript is intelligible, my main comments are larger, broad suggestions to improve the formatting (some focus) and readability of the manuscript.”

Reply to [Overall comment]: We thank the reviewer for appreciating the importance and contribution of our work. We mostly concurred with the reviewer’s suggestions and incorporated them into the revised manuscript accordingly. For details, see our replies to the reviewer’s specific comments. 

[C2-1]: Comment on “Rewriting the abstract”

 “The abstract needs to be rewritten and model the formatting of typical abstracts. It lacks details from the methods (participant information) and more specific details of the results, and their interpretation. The abstract is purpose or "introduction" heavy. I finished reading it unsure of what the project actually did and found.”

Reply to [C2-1]: As correctly pointed by the reviewer, we admit that the previous abstract failed to hit the balance between the “introductory”, “methodological”, and “results-summarizing” parts. In the revised abstract, we substantially cut the introductory part and provided more details about Methods and Results to help readers readily understand what our work is trying to address, how it does so, and what we eventually found (lines 20-47).

[C2-2]: Comment on “Format”: “The manuscript is generally well written. My concern is that the manuscript reads, and is formatted, closer to that of a thesis or dissertation paper, rather than a targeted journal article (e.g., how the introduction starts; many details in the methods could be supplemental – like power and sample size).”

Reply to [C2-2]: We agree with the reviewer’s points about the style of the previously submitted manuscript. As suggested, we deleted the starting part of Introduction, which sounds too general (lines 50-57), and presented two Methods sections ("power and sample size" and "De-confounding the CCA from the overall tendency of making extreme reports") as Supplementary Appendices in the revised manuscript. 

[C2-3]: Comment on “figure caption section”: “The Figure captions being included within the body of the manuscript might be allowed for the journal submission, but they would be better suited in a "Figure Caption" section if able.”

Reply to [C2-3]: We felt the same way with the reviewer that “The Figure captions being included within the body of the manuscript” is distractive. However, unfortunately, this is one of the format requirements imposed by PLOS ONE. We will appreciate your understanding on this matter.

[C2-4]: Comment on “the figures”: “As for the figures, reconsider what items are necessary and those that are not (e.g., Fig 1 – A, B, C, D and F do not seem necessary to me; Fig 5 - skewness, kurtosis etc is not needed here). Demographics can be tabled, and would be easier to understand. There should be more room dedicated to visualizing your data/results.”

Reply to [C2-4]: To incorporate the reviewer’s point, which we agree with, in the revised manuscript, we carefully reconsidered what figure panels are necessary for readers to understand the important points of our work and justified our decisions as follows. As for Fig 1, we decided to remove the old Fig 1A and present it as Table (Table 1 in the revised manuscript) because the demographic information can be more specific in a Table format. We also decided to remove Fig 1B, Fig 1C, and Fig 1F because the information depicted in those panels is sufficiently available in the text. However, we decided to keep the old Fig 1D (, which is Fig 1A in the revised manuscript) because we need this figure to illustrate how the average emotion responses to the VN stimuli were distributed in the affective space, which was necessary to address one of the comments made the other reviewer (see [C1-2-6] in above and our reply to that). As for Fig 5, we decided to keep the table of skewness and kurtosis values. This decision was made because we think those values would help readers appreciate how the emotion responses are distributed differently between the two groups, which is crucial for characterizing the CCA mode of variation (Polarized arousal responses in the EM-high group), by providing quantitative indices of distribution shapes. Another reason to juxtapose those values with their corresponding bins of arousal values is to allow readers to appreciate the fact that “Polarized arousal responses” are expressed in terms of skewness for the extreme-value bins (e.g., nr<-2, 2<nr) and terms of kurtosis for the intermediate-value bins. We will appreciate it if the reviewer understands our decisions - and our corresponding justifications - on this matter. 

[C2-5]: Comment on “application of the current work”: “One quick statement in the discussion about how the work could be utilized in populations like schizophrenia opens the door for a very important application that the current work should highlight more. Performing this work in mood and anxiety disorders is a natural next step that is not discussed in the future work section at the end of the discussion. It should be, and projected findings even suggested.”

Reply to [C2-5]: We thank the reviewer for bringing up an important point. To address this point, we reviewed the translational research on emotional deficits in schizophrenia and anxiety-and-mood disorders and attempted to suggest possible ways of applying our approach to those clinical populations in the revised manuscript (lines 893-904).

---

## [Decision Letter · Decision Letter 1]

28 Jan 2022

A robust multivariate structure of interindividual covariation between psychosocial characteristics and arousal responses to visual narratives

PONE-D-21-14011R1

Dear Dr. Lee,

We’re pleased to inform you that your manuscript has been judged scientifically suitable for publication and will be formally accepted for publication once it meets all outstanding technical requirements.

Kind regards,

Christos Papadelis, Ph.D.

Academic Editor

PLOS ONE

Additional Editor Comments (optional):

Reviewers' comments:

Reviewer's Responses to Questions

**Comments to the Author**

1. If the authors have adequately addressed your comments raised in a previous round of review and you feel that this manuscript is now acceptable for publication, you may indicate that here to bypass the “Comments to the Author” section, enter your conflict of interest statement in the “Confidential to Editor” section, and submit your "Accept" recommendation.

Reviewer #2: All comments have been addressed

2. Is the manuscript technically sound, and do the data support the conclusions?

Reviewer #2: Yes

3. Has the statistical analysis been performed appropriately and rigorously? 

Reviewer #2: Yes

4. Have the authors made all data underlying the findings in their manuscript fully available?

Reviewer #2: Yes

5. Is the manuscript presented in an intelligible fashion and written in standard English?

Reviewer #2: Yes

6. Review Comments to the Author

Reviewer #2: The authors were thoughtful and thorough in their revision of the manuscript. All of my comments have been addressed. I have no further comments, so I recommend the revision be accepted for publication.

7. PLOS authors have the option to publish the peer review history of their article (what does this mean?). If published, this will include your full peer review and any attached files.

Reviewer #2: No

---

## [Editor Report · Acceptance letter]

7 Feb 2022

PONE-D-21-14011R1 

A robust multivariate structure of interindividual covariation between psychosocial characteristics and arousal responses to visual narratives 

Dear Dr. Lee:

I'm pleased to inform you that your manuscript has been deemed suitable for publication in PLOS ONE. Congratulations! Your manuscript is now with our production department. 

Kind regards, 

on behalf of

Dr. Christos Papadelis 

Academic Editor

PLOS ONE